# Confounding Robust Meta-Reinforcement Learning: A Causal Approach

## Abstract

Meta-Reinforcement Learning (Meta-RL) focuses on training policies using data collected from a variety of diverse environments. This approach enables the policy to adapt to new settings with only a few training steps. While many Meta-RL methods have demonstrated success, they often rely on the assumption that unobserved confounders can be excluded *a priori*. This paper investigates robust Meta-RL in sequential decision-making, given confounded observational data collected across multiple heterogeneous environments. We introduce a novel augmentation procedure for standard Meta-RL algorithms (e.g., MAML), which employs partial identification methods to generate posterior counterfactual trajectories from candidate environments that align with the confounded observations. These counterfactual trajectories are then used to find a policy initialization that produces strong generalization performance in the target domain. Theoretical analysis reveals that our causal Meta-RL approach is guaranteed to yield a solution that minimizes generalization loss.

## 1 Introduction

The capability of rapid learning and generalization across heterogeneous domains is widely regarded as a hallmark of human intelligence. Meta-learning is a critical approach to exploring how to endow AI with the capacity for fast adaptation across different environments and learning tasks (Vilalta & Drissi, 2002). Among various paradigms of meta-learning, meta reinforcement learning (meta-RL) has emerged as a crucial and popular direction, as data efficiency is essential for achieving optimal decision-making policies in RL applications. Meta-RL improves data efficiency of RL-powered decision support systems by leveraging past data collected from interactions with different source domains to enable fast adaptation to new environments.

A variety of algorithms have been proposed for meta-RL, typically categorized by the form of inner-loop meta-parameterization: parameterized policy gradients (Finn et al., 2017; Raghu et al., 2019; Yoon et al., 2018), black box (Duan et al., 2017; Wang et al., 2016; Mishra et al., 2018), and task inference (Rakelly et al., 2019; Zintgraf et al., 2020; Humplik et al., 2019), to name a few. While these methods have achieved successes in practice, they rely on the crucial assumption that the actions observed in the data—along with the subsequent states and rewards they produce—are not simultaneously influenced by unobserved confounders. If this assumption is violated, the expected return of the policies becomes non-identifiable, meaning the effects cannot be determined from the available data. The following example illustrates such challenges in a simple meta-RL task.

**Example 1** (Challenges of Unmeasured Confounding)**.** Consider Windy Gridworlds described in Fig. 1a where the goal of the agent is to go through one of the three corridors and pick up the target key without touching the lava. For all tasks, their maps are similar except for the position and colors of the keys; each task is associated with a specific target key. At each time step $t$, the agent can take five possible actions $X_t$: `up`, `down`, `left`, `right`, or `stay-put`; there is also a wind $U_t$ blowing at each grid, following one of five directions: `east`, `south`, `west`, `north`, or `no-wind`. If the agent decides to move, its next state is shifted by both its action and the wind direction through the mechanism $S_{t+1} \leftarrow S_t + X_t + U_t$. Otherwise, the agent will stay put ($X_t \leftarrow$ `stay-put`) at its current position, regardless of the wind direction, i.e., $S_{t+1} \leftarrow S_t$. In general, the wind tempts to push the agent toward the lava; the closer the agent gets to the lava, the stronger the wind becomes.

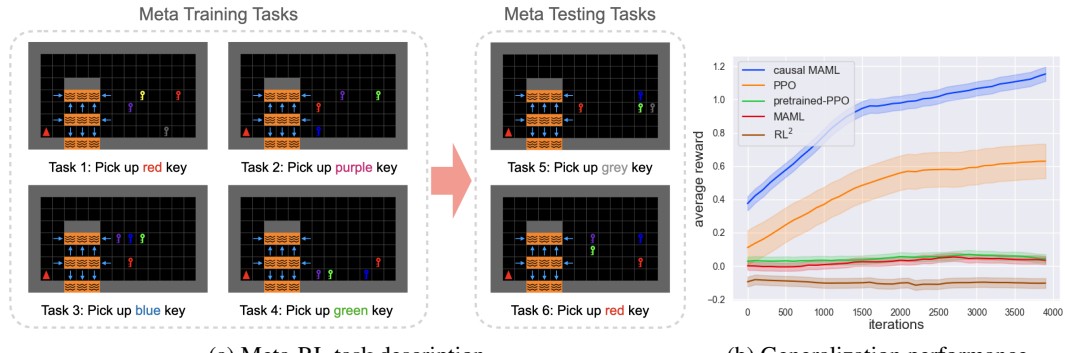

(a) Meta-RL task description    (b) Generalization performance

Figure 1: (a) Meta-RL tasks in a Windy Gridworld environment. Training and testing tasks are constructed by randomly generating key colors, key locations, and the target key. (b) few-shot adaptation performance comparing vanilla RL from scratch (PPO), pretrained RL (PRETRAINED-PPO), standard meta-learner (MAML), $RL^2$, and causally-enhanced meta-learner (CAUSAL-MAML).

The learning agent does not have access to the detailed system dynamics of each environment. Instead, it can observe an optimal behavioral agent that can sense the wind direction, operating in the training tasks described in Fig. 1a (left). After training, the learner will then be evaluated in the testing tasks described in Fig. 1a (left). In this meta-RL problem, the wind direction $U_t$ becomes an unobserved confounder affecting the observed action and state. We apply several meta-learning algorithms to this problem, including MAML (Finn et al., 2017), PPO (Schulman et al., 2017b), and $RL^2$(Duan et al., 2017) pretrained on observational data. For comparison, we also include a vanilla PPO without pretraining. Simulation results, shown in Fig. 1b, indicate that none of MAML, pretrained PPO, or $RL^2$ can outperform the vanilla PPO. We notice a significant gap between meta-learners and the vanilla one; the confounding bias in the observed data seems to affect the meta-learners' performance. ■

Recently, a growing body of literature has explored the nuanced interactions between causal inference theory and reinforcement learning to address data biases in the optimal decision-making under uncertainty, known as *Causal Reinforcement Learning (CRL)* (Bareinboim et al., 2024). Several algorithms have been proposed for various policy learning settings, including online learning (Bareinboim et al., 2015; Zhang & Bareinboim, 2017), off-policy learning (Kallus & Zhou, 2018; Namkoong et al., 2020; Etesami & Geiger, 2020; Zhang & Bareinboim, 2025), imitation learning (de Haan et al., 2019; Ruan et al., 2023; 2024), and curriculum learning (Li et al., 2025b), to name a few. Few works (Dasgupta et al., 2019b;a) have explored causal structure discovery and causal reasoning using meta-learning approaches. Despite these progresses, a systematic approach for applying meta-learning to sequential decision-making tasks in finite action and state spaces with the presence of unmeasured confounding is still missing. It is unclear how one can obtain a model initialization with reasonable generalization performance when the training data is contaminated with confounding bias and potential shifts occur in the system dynamics of the testing environment.

This paper aims to address a significant gap in the field by investigating robust meta-reinforcement learning (meta-RL) using confounded observational data gathered from various unknown Markov decision processes with similar yet distinct system dynamics. A key aspect of our approach is to employ partial causal identification, as discussed by (Balke & Pearl, 1994), alongside the representation of causal generative models introduced by (Zhang et al., 2022). More specifically, our contributions are summarized as follows. (1) We introduce a novel robust meta-RL method that leverages confounded observational data to predict non-identifiable system dynamics of the source domains while generating new counterfactual trajectories for training a meta-policy with enhanced adaptability across confounded environments. (2) We provide theoretical guarantees regarding the convergence of our method and detail the sample complexity necessary to obtain a good first-order stationary point approximation for the meta-RL policy. Finally, we validate our proposed algorithm through comprehensive simulations in synthetic RL environments. Due to space constraints, all proofs and detailed descriptions of the experimental setups can be found in the Appendix.

**Notations.** We use capital letters to denote random variables $(X)$, small letters for their values $(x)$, and calligraphic letters $\mathcal{X}$ for the domain of $X$. For an arbitrary set $\boldsymbol{X}$, let $|\boldsymbol{X}|$ be its cardinality. Fix indices $i, j \in \mathbb{N}$. Let $\boldsymbol{X}_{i:j}$ stand for a sequence of variables $\{X_i, X_{i+1}, \ldots, X_j\}$; We denote by $P(\boldsymbol{X})$ a probability distribution over variables $\boldsymbol{X}$, and will consistently use $P(\boldsymbol{x})$ as abbreviations for probabilities $P(\boldsymbol{X} = \boldsymbol{x})$. Finally, $\mathbb{1}_{\boldsymbol{X}=\boldsymbol{x}}$ is an indicator function that returns 1 if an event $\boldsymbol{X} = \boldsymbol{x}$ holds true; otherwise, it returns a constant 0.

## 2 META-REINFORCEMENT LEARNING WITH UNMEASURED CONFOUNDING

We will consider the sequential decision-making setting where the agent intervenes on a sequence of actions to optimize subsequent rewards. Throughout this paper, we will focus on a generalized family of confounded MDPs (Zhang & Bareinboim, 2016; Kallus & Zhou, 2020; Bennett et al., 2021) where the unobserved confounders are assumed away *a priori*, and the learner does not necessarily have the liberty to control how the behavioral policy generates the observational data.

**Definition 1.** A Confounded Markov Decision Process (CMDP) $\mathcal{M}$ is a tuple of $\langle \mathcal{S}, \mathcal{X}, \mathcal{Y}, \mathcal{U}, \mathcal{F}, P \rangle$ where (1) $\mathcal{S}, \mathcal{X}, \mathcal{Y}$ are, respectively, the spaces of observed states, actions, and rewards; (2) $\mathcal{U}$ is the space of unobserved exogenous noise; (3) $\mathcal{F}$ is a set consisting of the transition function $f_S : \mathcal{S} \times \mathcal{X} \times \mathcal{U} \mapsto \mathcal{S}$, behavioral policy $f_X : \mathcal{S} \times \mathcal{U} \mapsto \mathcal{X}$, and reward function $f_Y : \mathcal{S} \times \mathcal{X} \times \mathcal{U} \mapsto \mathcal{Y}$; (4) $P$ is an exogenous distribution over the domain $\mathcal{U}$.

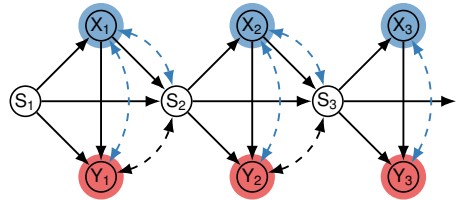

Throughout this paper, we will consider CMDPs with a finite horizon $H < \infty$; we consistently assume the action domain $\mathcal{X}$ and the state domain $\mathcal{S}$ to be discrete and finite; the reward domain $\mathcal{Y}$ is bounded in a real interval $[a, b] \subset \mathbb{R}$. A policy $\pi$ in a CMDP $\mathcal{M}$ is a decision rule $\pi(x_t \mid s_t)$ mapping from state to a distribution over action domain $\mathcal{X}$. An intervention $do(\pi)$ is an operation that replaces the behavioral policy $f_X$ in CMDP $\mathcal{M}$ with the policy $\pi$ (Pearl, 2000, Ch. 5). Let $\mathcal{M}_\pi$ be the submodel induced by

Figure 2: Causal diagram representing the data-generating mechanisms in a Confounded Markov Decision Process.

intervention $do(\pi)$. The interventional distribution $P_\pi(\bar{\boldsymbol{X}}_{1:H}, \bar{\boldsymbol{S}}_{1:H}, \bar{\boldsymbol{Y}}_{1:H})$ is defined as the joint distribution over observed variables in thus post-interventional submodel $\mathcal{M}_\pi$,

$$P_\pi(\bar{\boldsymbol{x}}_{1:H}, \bar{\boldsymbol{s}}_{1:H}, \bar{\boldsymbol{y}}_{1:H}) = P(s_1) \prod_{t=1}^{H} \left( \pi(x_t \mid s_t) \mathcal{T}(s_t, x_t, s_{t+1}) \mathcal{R}(s_t, x_t, y_t) \right) \tag{1}$$

where the transition distribution $\mathcal{T}$ and the reward distribution $\mathcal{R}$ are given by, for $t = 1, \ldots, H$,

$$\mathcal{T}(s_t, x_t, s_{t+1}) = \int_{\mathcal{U}} \mathbb{1}_{s_{t+1}=f_S(s_t, x_t, u_t)} P(u_t), \quad \mathcal{R}(s_t, x_t, y_t) = \int_{\mathcal{U}} \mathbb{1}_{y_t=f_Y(s_t, x_t, u_t)} P(u_t). \tag{2}$$

For convenience, we write the reward function $\mathcal{R}(s, x)$ as the expected value $\sum_y y \mathcal{R}(s, x, y)$. A realization of states and actions is called a trajectory and can be written as $\tau = (\bar{\boldsymbol{x}}_{1:H}, \bar{\boldsymbol{s}}_{1:H}, \bar{\boldsymbol{y}}_{1:H})$.

A common objective for an RL agent is to optimize its cumulative return $J_\pi = \mathbb{E}_\pi \left[ \sum_{t=1}^{H} \gamma^{t-1} Y_t \right]$ where $0 \leq \gamma \leq 1$ is the discount factor. When detailed parametrizations of the underlying distribution and function are provided, there exist standard planning methods to compute the optimal policy (Bellman, 1966; Sutton & Barto, 1998). However, in many practical scenarios, the detailed knowledge of the environments is often not fully available. In this paper, we consider learning settings where the agent has access to the observational data in CMDPs, generated by demonstrators following behavioral policies. Specifically, for every time step $t = 1, \ldots, H$, the environment first draws an exogenous noise $U_t$ from the distribution $P(\mathcal{U})$; the demonstrator then performs an action $X_t \leftarrow f_X(S_t, U_t)$ following the behavioral policy $f_X$, receives a subsequent reward $Y_t \leftarrow r_t(S_t, X_t, U_t)$, and moves to the next state $S_{t+1} \leftarrow f_S(S_t, X_t, U_t)$. The observed trajectories are summarized as the observational distribution $P(\bar{\boldsymbol{X}}_{1:H}, \bar{\boldsymbol{S}}_{1:H}, \bar{\boldsymbol{Y}}_{1:H})$,

$$P(\bar{\boldsymbol{x}}_{1:H}, \bar{\boldsymbol{s}}_{1:H}, \bar{\boldsymbol{y}}_{1:H}) = P(s_1) \prod_{t=1}^{H} \left( \int_{\mathcal{U}} \mathbb{1}_{s_{t+1}=f_S(s_t, x_t, u_t)} \mathbb{1}_{x_t=f_X(s_t, u_t)} \mathbb{1}_{y_h=f_Y(s_t, x_t, u_t)} P(u_t) \right). \tag{3}$$

Fig. 2 shows the causal diagram $\mathcal{G}$ (Bareinboim et al., 2022) describing the generative process of the observational data in CMDPs, where nodes represent observed variables $X_t, S_t, Y_t$, and arrows represent the functional relationships $f_X, f_S, f_Y$ among them. Exogenous variables $U_t$ are often not explicitly shown; bi-directed arrows $X_t \longleftrightarrow Y_t$ and $X_t \longleftrightarrow S_{t+1}$ (highlighted in blue) indicate the presence of an unobserved confounder (UC) $U_t$ affecting the action, state, and reward simultaneously. The presence of these unobserved confounders violates the conditions of no unmeasured confounding (Robbins, 1985; Bareinboim et al., 2024), leading to possible challenges for various policy learning tasks, including meta-RL (Finn et al., 2017), which will be the focus of the remainder of this paper.

**Meta-Reinforcement Learning.** Let $\mathcal{B} = \{\mathcal{M}_i\}_{i=1}^B$ be the set of CMDPs representing different RL tasks. We assume these CMDPs are drawn from a distribution $\rho$ (which Nature will draw samples from). The detailed parametrizations of exogenous distribution $P_i$ and structural functions $\mathcal{F}_i$ for these CMDPs $\mathcal{M}_i$ generally differ from one another. We will consistently use $\mathcal{D}_{\text{obs}}^i$ to denote trajectories collected passively observing a demonstrator operating in the model $\mathcal{M}_i$, following the observational distribution of Eq. (3). Similarly, we use $\mathcal{D}_{\text{exp}}^i$ to denote the experimental trajectories collected from performing interventions $\text{do}(\pi_i)$ in the model $\mathcal{M}_i$ following some policies $\pi_i$, i.e., $\mathcal{D}_{\text{exp}}^i$ are drawn from the interventional distribution of Eq. (1).

To demonstrate our general data augmentation technique, we apply it to a well-known meta-RL method, MAML (Finn et al., 2017). The goal of MAML is to learn a policy $\pi$ that peforms well as an initialization for learning a new unseen task $\mathcal{M}_i$ when the learner has a budget for running a few steps of gradient descent. To search over the space of all policies, we assume these policies are parametrized with $\theta \in \mathbb{R}^d$. We denote the policy corresponding to parameter $\theta$ by $\pi(\cdot; \theta)$ and the expected return corresponding to this policy $\pi(\cdot; \theta)$ in a model $\mathcal{M}_i$ by $J_i(\theta)$. For simplicity, we focus on finding an initialization $\theta$ such that, after observing a new CMDP $\mathcal{M}_i$, one gradient step would lead to a good approximation for the minimizer of $J_i(\theta)$. We can formulate this learning goal as follows

$$\max_{\theta} F(\theta) := \mathbb{E}_{\mathcal{M}_i \sim \rho} \left[ J_i \left( \theta + \alpha \nabla J_i(\theta) \right) \right], \tag{4}$$

where the step size $\alpha$ is a hyper-parameter that controls the magnitude of the gradient ascent update. In other words, the optimal solution of Eq. (4) would perform well in expectation when the learner is deployed to a CMDP task and looks at the output after running a single step of gradient descent.

In practice, however, since the detailed system dynamics of the target CMDP $\mathcal{M}_i$ are unknown, one must estimate the policy gradient $\nabla J_i(\theta)$ from empirical samples collected from the environment. Unbiased estimation methods have been proposed (Finn et al., 2017; Fallah et al., 2020) to approximate the gradient when the learner could directly intervene in the environment. Specifically, the learner will intervene in the CMDP $\mathcal{M}_i$, collect a batch of experimental data $\mathcal{D}_{\text{exp}}^i$, evaluate the stochastic gradient $\tilde{\nabla} J_i(\theta, \mathcal{D}_{\text{exp}}^i)$ from the batch, and solve for the optimal solution $\theta$ of Eq. (4) by replacing the gradient $\nabla J_i(\theta)$ with $\tilde{\nabla} J_i(\theta, \mathcal{D}_{\text{exp}}^i)$. When $\tilde{\nabla} J_i(\theta, \mathcal{D}_{\text{exp}}^i)$ is an unbiased estimator, this meta-RL approach has demonstrated success and achieved an optimal initialization point $\theta^*$.

However, challenges could arise when the agent does not have access to directly intervene in the task $\mathcal{M}_i$. Without realizing the discrepancy between the observational $\mathcal{D}_{\text{obs}}^i$ and experimental data $\mathcal{D}_{\text{exp}}^i$, a naive learner might use $\mathcal{D}_{\text{obs}}^i$ as if it were $\mathcal{D}_{\text{exp}}^i$, and proceed with the original MAML method. This procedure leads to the following optimization program:

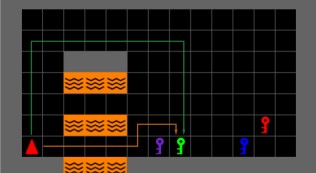

$$\max_{\theta} \tilde{F}(\theta) = \mathbb{E}_{\mathcal{M}_i \sim \rho} \left[ \mathbb{E}_{\mathcal{D}_{\text{obs}}^i} \left[ J_i \left( \theta + \alpha \tilde{\nabla} J_i(\theta, \mathcal{D}_{\text{obs}}^i) \right) \right] \right]. \tag{5}$$

Figure 3: Comparing two possible routes (long and short) to reach the target green key.

Among the above quantities, $\tilde{\nabla} J_i(\theta, \mathcal{D}_{\text{obs}}^i)$ is the stochastic gradient evaluated from the observational data $\mathcal{D}_{\text{obs}}^i$. Generally, when the unobserved confounding exists, the underlying system dynamics are underdetermined (i.e., non-identifiable) from the observational data (Kallus & Zhou, 2018; Zhang & Bareinboim, 2025). Consequently, the stochastic gradient $\tilde{\nabla} J_i(\theta, \mathcal{D}_{\text{obs}}^i)$ is no longer an unbiased estimate of $\nabla J_i(\theta)$, and solving the optimization in Eq. (5) yields a solution $\theta$ with sub-optimal behavior.

**Example 2** (Windy Gridworlds continued)**.** Consider the meta-reinforcement learning task of windy gridworlds described in Fig. 1a. In this scenario, the wind direction $U_t$ serves as an unobserved

confounder that influences the observed action $X_t$, the subsequent reward $Y_t$, and the next state $S_{t+1}$. This introduces spurious correlations in the observational data, causing some trajectories to appear associated with higher returns. For example, Fig. 3 illustrates two observed trajectories leading to the target green key. The shorter orange route is risky, as it requires navigating a narrow passage between lava tiles. The demonstrator, able to sense the wind direction, can stop when pushed toward the lava and thus consistently take the short route to reach the key. However, the learner cannot sense the wind and cannot choose the right moment to stop. If the learner naively updates its policy using the stochastic gradient $\tilde{\nabla} J_i(\theta, \mathcal{D}^i_{obs})$ derived from the observational data, it will not accurately recover the actual gradient $\nabla J_i(\theta)$. Instead, it will overestimate the value of the risky short route trajectories, leading to sub-optimal performance. In contrast, the learner should consider taking the longer but safer upper passage, which is more reliable even in windy conditions. ∎

To better highlight the difference between the optimal policy initialization for meta-RL in Eq. (4) and the biased solution obtained by naively applying standard MAML in Eq. (5) with confounded observations, we consider an example with three equally likely CMDPs $\mathcal{M}_1, \mathcal{M}_2, \mathcal{M}_3$; see Fig. 4. For each sampled CMDP $\mathcal{M}_i$, the dashed shade represents the equivalence class of environments $\tilde{M}_i$ compatible with the same observational data. When unmeasured confounding exists, one cannot distinguish between the actual task $\mathcal{M}_i$ and the other task $\tilde{\mathcal{M}}_i$, and these models could have significantly different system dynamics. If one is not aware of this difference and naively applies MAML gradient update using confounded observations, the algorithm will converge to the alternative task $\tilde{\mathcal{M}}_i$ in the equivalence. When the confounding bias is significant and $\tilde{\mathcal{M}}_i$ deviates from the actual task $\mathcal{M}_i$, the obtained solution $\tilde{\theta}$ could deviate from the optimal $\theta$ and fail to generalize to all environments.

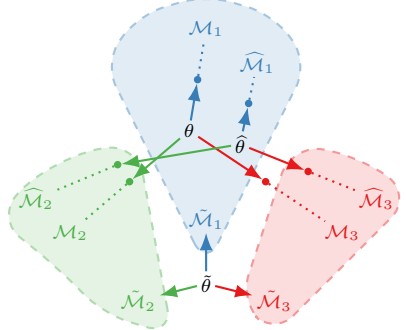

Figure 4: Comparing the optimal solution $\theta$ of Eq. (4) and solutions obtained by naive meta-RL $\tilde{\theta}$ (Eq. (5)) and the causally enhanced approach $\hat{\theta}$ (Eq. (6)).

## 3 CONFOUNDING ROBUST META-REINFORCEMENT LEARNING

A natural question arising at this point is how to perform robust meta-RL in the face of unmeasured confounding in the observational data. Our analysis so far seems to suggest that when the no-unmeasured-confounding condition does not hold, it is infeasible to obtain an unbiased stochastic gradient for the policy update, preventing the recovery of the optimal meta-policy in Eq. (4). For the remainder of this paper, we will show that this is not the case by proposing a novel confounding-robust meta-RL algorithm leveraging counterfactual reasoning and providing theoretical guarantees that it recovers the optimal meta-policy under some common conditions.

Note that CMDP tasks $\mathcal{M}_i$ are drawn from a prior distribution $\rho$. Our discussion begins with a meta-RL approach assuming access to an oracle capable of sampling the posterior tasks $\widehat{\mathcal{M}}_i \sim \rho(\mathcal{M} \mid \mathcal{D}^i_{\text{obs}})$ conditioned on the observational data $\mathcal{D}^i_{\text{obs}}$. We will then relax this assumption by providing a practical Monte-Carlo approach to sample the posterior distribution. Specifically, after observing a CMDP task $\mathcal{M}_i$ and receiving the observational data $\mathcal{D}^i_{\text{obs}}$, instead of evaluating the gradient $\nabla J_i(\theta)$ from confounded observations, our causal learner will sample an alternative model $\widehat{\mathcal{M}}_i$ compatible with the same observations from the oracle $\rho(\mathcal{M} \mid \mathcal{D}^i_{\text{obs}})$. The causal meta-learner will then interact with this posterior model $\widehat{\mathcal{M}}_i$ and collect the subsequent experimental data $\widehat{\mathcal{D}}^i_{\text{exp}}$. Finally, the causal learner performs the stochastic gradient update $\widehat{\nabla} J_i(\theta, \widehat{\mathcal{D}}^i_{\text{exp}})$ using the posterior experimental data. This augmented meta-RL procedure could be formalized as the following optimization program:

$$\max_{\theta} \widehat{F}(\theta) := \mathbb{E}_{\mathcal{M}_i \sim \rho} \left[ \mathbb{E}_{\mathcal{D}^i_{\text{obs}}} \left[ \mathbb{E}_{\widehat{\mathcal{D}}^i_{\text{exp}}} \left[ J_i \left( \theta + \alpha \widehat{\nabla} J_i(\theta, \widehat{\mathcal{D}}^i_{\text{exp}}) \right) \right] \right] \right]. \qquad (6)$$

In the above equation, computing the posterior experimental data $\widehat{\mathcal{D}}^i_{\text{exp}}$ conditioned on the observational trajectories $\mathcal{D}^i_{\text{obs}}$ can be seen as performing a counterfactual query. That is, *"given the observed trajectories (collected from the demonstrator), what would the outcome be had I personally taken the same route as the observed one (or exploring an alternative route)?"* Henceforth, we will consistently

refer to this augmentation step as the *counterfactual bootstrap*. We will later show that this bootstrapping step effectively mitigates the influence of unobserved confounders, enabling the learner to obtain the optimal policy initialization. Fig. 4 illustrates this intuition by comparing the solution $\widehat{\theta}$ of Eq. (6) to the optimal solution of Eq. (4). Here, $\widehat{\theta}$ is a meta-policy computed using the counterfactual CMDPs drawn from the oracle $\widehat{\mathcal{M}}_i \sim \rho(\mathcal{M} \mid \mathcal{D}_{\text{obs}}^i)$. Since the oracle provides access to the posterior over all tasks conditioned on observed trajectories, the solution $\widehat{\theta}$ is a consistent estimate of the optimal solution in expectation, thereby leading to a reasonable generalization performance.

**Counterfactual Bootstrap.** The causal meta-reinforcement learning (meta-RL) method discussed earlier depends on having oracle access to the posterior distribution $\rho(\mathcal{M}_i \mid \mathcal{D}_{\text{obs}}^i)$, which is conditioned on the confounded observations. However, evaluating this posterior can be difficult in practice because we lack detailed information about the prior distribution $\rho(\mathcal{M})$ over potential tasks. One possible solution is to define a non-informative prior $\widehat{\rho}$ to serve as an approximation of the actual prior $\rho$. However, constructing such a prior $\widehat{\rho}$ is complicated, as we do not know the specific parametric forms of the distribution $P$ and the structural functions $\mathcal{F}$ for the underlying CMDPs. To address this challenge, we will utilize a parametric family of canonical causal models introduced by (Zhang et al., 2022), which limits the cardinality of the latent exogenous domain based on the cardinality of the observed state-action space. Formally, the canonical parameterization of CMDPs is provided as follows.

**Definition 2.** A canonical CMDP $\mathcal{M}$ is a CMDP $\langle \mathcal{S}, \mathcal{X}, \mathcal{Y}, \mathcal{U}, \mathcal{F}, P \rangle$ where its the cardinality of the exogenous domain $\mathcal{U}$ is bounded by $|\mathcal{U}| \leq 2(|\mathcal{S} \times \mathcal{X}| + |\mathcal{S} \times \mathcal{X} \times \mathcal{S}| + |\mathcal{S} \times \mathcal{X} \times \mathcal{Y}|)$.

For a canonical CMDP, the latent cardinality of the exogenous domain is bounded by a linear function of the cardinality of the observed state-action space. For standard CMDPs with discrete states and actions, the latent exogenous domain is also discrete and finite.[1] A critical property of canonical causal models is that they preserve the values of all the observational and interventional distributions defined by the original, unrestricted causal models using only a finite number of latent states. The following corollary follows immediately from (Zhang et al., 2022, Theorem 2.4).

**Corollary 1.** *For an arbitrary CMDP $\mathcal{M}$, there exists a canonical CMDP $\mathcal{N}$ such that for any finite horizon $H < \infty$ and any policy $\pi$, $P(\bar{\boldsymbol{x}}_{1:H}, \bar{\boldsymbol{s}}_{1:H}, \bar{\boldsymbol{y}}_{1:H}; \mathcal{M}) = P(\bar{\boldsymbol{x}}_{1:H}, \bar{\boldsymbol{s}}_{1:H}, \bar{\boldsymbol{y}}_{1:H}; \mathcal{N})$ and $P_\pi(\bar{\boldsymbol{x}}_{1:H}, \bar{\boldsymbol{s}}_{1:H}, \bar{\boldsymbol{y}}_{1:H}; \mathcal{M}) = P_\pi(\bar{\boldsymbol{x}}_{1:H}, \bar{\boldsymbol{s}}_{1:H}, \bar{\boldsymbol{y}}_{1:H}; \mathcal{N})$.*

Corol. 1 implies that for meta-RL tasks from the observational data over discrete domains, one could assume the latent states of the underlying CMDPs to be discrete and finite without loss of generality. This latent space reduction simplifies the construction of the approximate prior $\widehat{\rho}$. Specifically, we will follow the procedure of (Zhang et al., 2022) and assign a Dirichlet prior over the exogenous probabilities $P(\mathcal{U})$; structural functions $\mathcal{F}$ are uniformly drawn from a finite set of functional mappings between discrete domains. Provided with the prior $\widehat{\rho}(\mathcal{M})$ over CMDP tasks and observed trajectories $\mathcal{D}_{\text{obs}}^i$ in a model $\mathcal{M}_i$, there exist general Monte-Carlo Markov Chain algorithms to sample posterior tasks $\widehat{\rho}(\mathcal{M}_i \mid \mathcal{D}_{\text{obs}}^i)$, including Gibbs sampling (Gelfand & Smith, 1990) and Hamiltonian Monte Carlo (HMC) (Duane et al., 1987).

**Causal MAML.** We are now ready to introduce our general data augmentation technique applied to MAML, called CAUSAL-MAML, for confounded observations. Details are described in Alg. 1. Similar to many gradient-based model agnostic meta-learning methods (Finn et al., 2017; Fallah et al., 2020; 2021), its training procedures contain an inner loop and an outer loop. More specifically, at Line 3, Nature (e.g., a system designer) selects a collection of source meta-training CMDP tasks $\mathcal{B} = \{\mathcal{M}_i\}$ following the distribution $\rho$. For every CMDP $\mathcal{M}_i$ in the inner training loop, the learner observes its trajectories (generated by a demonstrator) and obtains the observational data $\mathcal{D}_{\text{obs}}^i$ (Line 5). It then constructs an approximate posterior $\widehat{\rho}(\mathcal{M} \mid \mathcal{D}_{\text{obs}}^i)$ and draws an alternative environment $\widehat{\mathcal{M}}_i$ from the posterior, following the counterfactual bootstrap procedure described previously. The learner simulates interventions following the current policy estimate $\pi(\cdot \mid \cdot; \theta)$ in the sampled CMDP $\widehat{\mathcal{M}}_i$ and collects experimental trajectories $\widehat{\mathcal{D}}_{\text{exp,in}}^i$ (Line 7). It then computes the inner stochastic gradient

---

[1]For continuous rewards $Y_t$ bounded in a compact domain $\mathcal{Y}$, one could always represent their first moments (e.g., reward function $\mathcal{R}(s_t, x_t)$) using a binary Bernoulli distribution (Agrawal & Goyal, 2012). The reward domain $\mathcal{Y}$ could be further discretized to represent higher moments.

---

**Algorithm 1:** CAUSAL-MAML

**1 Require:** Initial parameter $\theta$, an approximate prior over CMDPs $\widehat{\rho}(\mathcal{M})$

**2 while** not done **do**

**3**     Nature samples a batch of CMDP tasks $\mathcal{B} = \{\mathcal{M}_i\}_{i=1}^{B}$ from distribution $\rho(\mathcal{M})$

**4**     **for** all task $\mathcal{M}_i \in \mathcal{B}$ **do**

**5**        Sample observation trajectories $\mathcal{D}_{\text{obs}}^i$ in environment $\mathcal{M}_i$

**6**        Sample a new environment $\widehat{\mathcal{M}}_i$ from the posterior $\widehat{\rho}(\mathcal{M} \mid \mathcal{D}_{\text{obs}}^i)$

**7**        Sample experimental trajectories $\widehat{\mathcal{D}}_{\text{exp,in}}^i$ using agent policy $\pi(\cdot \mid \cdot; \theta)$ in environment $\widehat{\mathcal{M}}_i$

**8**        Compute inner gradient $\widehat{\nabla}_\theta J_i(\theta, \widehat{\mathcal{D}}_{\text{exp,in}}^i)$ using dataset $\widehat{\mathcal{D}}_{\text{exp,in}}^i$ following Eq. (7)

**9**        Set adapted parameter $\theta_i = \theta + \alpha \widehat{\nabla}_\theta J_i(\theta, \widehat{\mathcal{D}}_{\text{exp,in}}^i)$

**10**       Sample experimental dataset $\mathcal{D}_{\text{exp,o}}^i$ using adapted policy $\pi(\cdot \mid \cdot; \theta_i)$ in environment $\widehat{\mathcal{M}}_i$

**11**     **end**

**12**     Update parameter $\theta \leftarrow \theta + \beta \widehat{\nabla}_\theta F(\theta)$ following Eq. (8)

**13 end**

---

$\widehat{\nabla}_\theta J_i(\theta, \widehat{\mathcal{D}}_{\text{exp,in}}^i)$ using the collected experimental trajectories. Formally, given finite experimental trajectories $\widehat{\mathcal{D}}_{\text{exp}}$, we define the stochastic gradient $\widehat{\nabla}_\theta J_i(\theta, \widehat{\mathcal{D}})$ as follows:

$$\widehat{\nabla}_\theta J_i(\theta, \widehat{\mathcal{D}}) = \frac{1}{|\widehat{\mathcal{D}}|} \sum_{\tau \in \widehat{\mathcal{D}}} \sum_{t=0}^{H} \nabla_\theta \log \pi(x_t \mid s_t; \theta) \Psi_t, \quad \text{where} \quad \Psi_t = \sum_{t'=t}^{H} \gamma^{t'} \mathcal{R}_i(s_{t'}, x_{t'}). \quad (7)$$

At Lines 9-10, the learner updates the parameter $\theta_i$ of an adapted policy $\pi(\cdot \mid \cdot; \theta_i)$ and uses this policy to subsequently interact with the sampled CMDP $\widehat{\mathcal{M}}_i$ to generate outer-loop experimental trajectories $\widehat{\mathcal{D}}_{\text{exp,o}}^i$. After completing the inner training loop for every source task, the learner finally enters the outer-loop update and adjusts the parameter $\theta$ using the gradient of meta-RL objective function $\widehat{\nabla}_\theta F(\theta)$ evaluated at the adapted parameter $\theta_i$ and the outer-loop trajectories $\widehat{\mathcal{D}}_{\text{exp,o}}^i$. Formally, the stochastic gradient of the meta-objective function is defined as follows:[2]

$$\widehat{\nabla}_\theta F(\theta) = \frac{1}{|\mathcal{B}|} \sum_{i \in \mathcal{B}} \Bigg( \Big( I + \alpha \widehat{\nabla}_\theta^2 J_i(\theta, \widehat{\mathcal{D}}_{\text{exp,in}}^i) \Big) \widehat{\nabla}_\theta J_i \Big( \theta_i, \widehat{\mathcal{D}}_{\text{exp,o}}^i \Big)$$

$$+ \widehat{J}_i \Big( \theta_i, \widehat{\mathcal{D}}_{\text{exp,o}}^i \Big) \sum_{\tau \in \widehat{\mathcal{D}}_{\text{exp,in}}^i} \sum_{t=0}^{H} \nabla_\theta \log \pi(x_t \mid s_t; \theta) \Bigg). \quad (8)$$

Among quantites in the above equation, $I$ is an identity matrix; $\widehat{J}_i(\theta_i, \widehat{\mathcal{D}}_{\text{exp,o}}^i)$ is the empirical mean estimate of the expected return for a policy $\pi(\cdot \mid \cdot; \theta_i)$ evaluated from outer-loop trajectories $\widehat{\mathcal{D}}_{\text{exp,o}}^i$. $\widehat{\nabla}_\theta^2 J_i(\theta, \widehat{\mathcal{D}})$ is policy Hessian estimate for sampled CMDP $\widehat{\mathcal{M}}_i$ defined as

$$\widehat{\nabla}_\theta^2 J_i(\theta, \widehat{\mathcal{D}}) = \frac{1}{|\widehat{\mathcal{D}}|} \sum_{\tau \in \widehat{\mathcal{D}}} \Bigg( \Big( \sum_{t=0}^{H} \nabla_\theta \log \pi(x_t \mid s_t; \theta) \Psi_t \Big) \times \nabla_\theta \log p_i(\tau; \theta)$$

$$+ \sum_{t=0}^{H} \nabla_\theta^2 \log \pi(x_t \mid s_t; \theta) \Psi_t \Bigg) \quad (9)$$

with the interventional probability $p_i(\tau; \theta) = P_{\pi(\cdot \mid \cdot; \theta)}(\tau)$. It can be verified that if all the gradients and Hessians in the outer-loop update were exact, then the outcome of the update would be equivalent to the outcome of the gradient ascent update for the objective function $\widehat{F}(\theta)$ (Fallah et al., 2021).

---

[2]For simplicity, we assume that all experimental trajectories $\widehat{\mathcal{D}}_{\text{exp,in}}^i$ and $\widehat{\mathcal{D}}_{\text{exp,o}}^i$ have the same size $D$.

## 3.1 Convergence of Causal MAML

For the remainder of this section, we will analyze the asymptotic properties of our proposed CAUSAL-MAML algorithm and provide theoretical guarantees for the computational complexity of its convergence. Our discussion begins with introducing some necessary conditions on the smoothness of the hypothesis class containing the candidate policy networks.

**Assumption 1.** The gradient and Hessian of logarithmic policy are bounded; that is, there exist constants $G, L \in \mathbb{R}$ such that, for any state $s \in \mathcal{S}$, action $x \in \mathcal{X}$, and parameter $\theta \in \mathbb{R}^d$, we have $\|\nabla_\theta \log \pi_\theta(x \mid s; \theta)\| \leq G$ and $\|\nabla_\theta^2 \log \pi(x \mid s; \theta)\| \leq L$.

**Assumption 2.** The Hessian of logarithmic policy is $K$-Lipschitz continuous; that is, there exists a real constant $K > 0$ such that for all parameters $\theta_1, \theta_2 \in \mathbb{R}^d$, state $s \in \mathcal{S}$ and action $x \in \mathcal{X}$, we have $\|\nabla_\theta^2 \log \pi(x \mid s; \theta_1) - \nabla_\theta^2 \log \pi(x \mid s; \theta_2)\| \leq K\|\theta_1 - \theta_2\|$.

Assumption 1 states that the gradient and Hessian of the logarithmic policy distribution are bounded, and Assumption 2 implies that the Hessian of the logarithmic policy distribution is Lipschitz continuous. In practice, these assumptions generally hold for some common choices of hypothesis class of candidate policies, including neural networks with softmax layers (Bridle, 1990) and smooth activation functions (Dugas et al., 2000).

In practice, the meta-RL problem of Fig. 4 is generally non-convex. Due to this reason, we will focus on finding a policy initialization that satisfies the first-order optimality condition. Formally, a solution $\theta_\epsilon \in \mathbb{R}^d$ is called an $\epsilon$-approximate first-order stationary point ($\epsilon$-FOSP), if it satisfies $\|\nabla F(\theta_\epsilon)\| \leq \epsilon$, i.e., it approximates a local optimum of the meta-objective function. Our following result establishes the convergence of the proposed causal meta-learner.

**Theorem 1.** *Consider the case that $\alpha \in (0, 1/\eta_H]$ and $\beta \in (0, 1/L_H]$. For any $\epsilon \in (0, 1)$, CAUSAL-MAML finds a solution $\theta_\epsilon$ satisfying $E[\|\nabla_\theta F(\theta_\epsilon)\|^2] \leq 2L_G^2 L_H \beta B^{-1} D^{-1} + \epsilon^2$, after running at most for $\mathcal{O}(1)(b-a)(1-\gamma)^{-1}\beta^{-1} \min(\epsilon^{-2}, BDL_G^{-2}L_H^{-1}\beta^{-1}/2)$ iterations.*

Thm. 1 implies that our proposed causal meta-learner is guaranteed to find a local-optimum solution for the policy initialization of Fig. 4 with a sufficient number of iterations and trajectories. It also allows us to characterize the computational complexity of CAUSAL-MAML for finding an $\epsilon$-FOSP solution. Fix an error rate $\epsilon > 0$. The convergence condition of Thm. 1 implies two possible settings: (1) when $\beta = 1/L_G$, our CAUSAL-MAML requires at least $\mathcal{O}(\epsilon^{-2})$ iterations, with a total number of $\epsilon^{-2}$ trajectories per iteration to reach an $\epsilon$-FOSP solution; and (2) $\beta = \epsilon^{-2}$, CAUSAL-MAML requires at least a total number of $\mathcal{O}(\epsilon^{-4})$ iterations, with $\mathcal{O}(1)$ trajectories per iteration. In both cases, the total number of stochastic gradient evaluations is $\mathcal{O}(\epsilon^{-4})$.

## 4 Experiments

In this section, we validate our confounding robust meta-RL approach in the Windy Gridworlds (Li et al., 2025a; Zhang & Bareinboim, 2025), which is adapted from the Minigrid environment (Chevalier-Boisvert et al., 2023). In these environments, the agent is required to navigate around impassable terrain (e.g., walls and lava) and interact with specific objects (e.g., keys and doors). Winds are introduced in the passages between lava as unobserved confounders, affecting the agent's movements. For each task, interactive objects are assigned colors from a set of four; one color is designated as the unique target, while the remaining three serve as distractors. The source domain uses the palette {red, green, blue, purple}, while the target domain expands this palette with two additional colors, {yellow, gray}. We evaluate our approach on three meta-RL tasks: Pick-Up-Key (Experiment 1), Go-To-Door (Experiment 2), and Go-To-Goal (Experiment 3). Each environment contains four tasks in the source domain and two tasks in the target domain.

We assess the performance of algorithms by their ability to adapt to target tasks, specifically, quantified by the accumulated reward

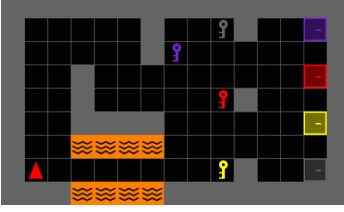

(a) Go-To-Door

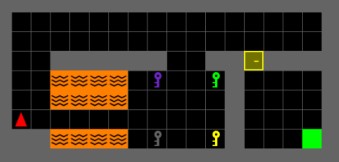

(b) Go-To-Goal

Figure 5: Meta-RL tasks in the windy Gridworld environments.

obtained during adaptation. For all baselines, the meta-policy is adapted to the target task using Proximal Policy Optimization (PPO) (Schulman et al., 2017a). Our method is compared to three baselines: (a) PPO: random initialization of meta-policy parameters; (b) MAML: training the meta-policy on demonstrator data using MAML; (c) $RL^2$: training the meta-policy on demonstrator data using $RL^2$, and (d) PRETRAINED-PPO: pretraining the meta-policy on demonstrator data. Implementation details for benchmark algorithms are provided in Appendix C.1. Furthermore, we present a comparison between pretraining over counterfactual environments generated from demonstrator data and our proposed method in Appendix C.2.

The policy model for the actor-critic network consists of a two-headed multilayer perceptron (MLP). Both the actor and critic heads share a fully connected layer with $64$ units, and each head features a single hidden layer MLP with $64$ hidden units. During the meta-training stage, we train the model for $300$ iterations. In the adaptation stage, we select five tasks from the target domain, train for $4,000$ iterations, and calculate the average accumulated reward across the tasks. Each iteration uses $512$ frames from the environments.

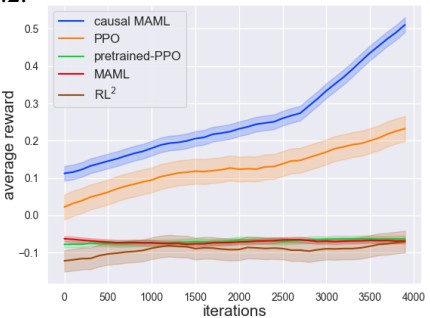

(a) Go-To-Door

**Experiment 1.** In the first experiment, the agent is trained to navigate in a $15 \times 9$ grid and to find the key of the target color. Details of this meta-RL task have been described in Fig. 1a. Keys are uniformly generated within the subgrid $\{(c, r) \mid 7 \leq c \leq 13, 4 \leq r \leq 7\}$. The wind distribution in the passages between lava is 0.1, 0.35, 0.1, 0.35, 0.1 for rightward, downward, leftward, upward, and staying in place, respectively. In other cells, the distribution is 0.01, 0.01, 0.01, 0.01, 0.96, indicating negligible wind effects. If the agent enters lava, a negative reward is received, while approaching the target key yields a positive reward. Simulation results in Fig. 1b suggest that confounding robust Meta-RL adapts more quickly and exhibits lower variance during adaptation compared to PPO. MAML, PRETRAINED-PPO, and $RL^2$ (Duan et al., 2016) fail to learn useful information from confounded data.

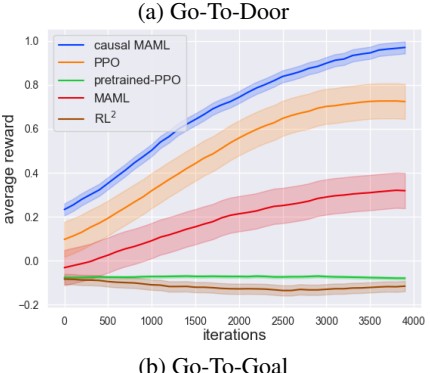

(b) Go-To-Goal

Figure 6: Cumulative returns comparing PPO from scratch, PRETRAINED PPO, standard MAML, and proposed CAUSAL-MAML.

**Experiment 2.** In the second experiment, the agent is required to pick up the target color key and open the corresponding door in a $15 \times 9$ grid. The environment is illustrated in Fig. 5a. Key locations are uniformly generated from the set $\{(7, 2), (9, 1), (9, 4), (9, 7)\}$, and door locations are uniformly generated from the set $\{(13, 1), (13, 3), (13, 5), (13, 7)\}$. The wind distribution in the lava passage and other cells is identical to the description in Experiment 1. Entering lava produces a negative reward. Before obtaining the target key, approaching it yields a positive reward; after acquiring the target key, approaching the corresponding door provides a positive reward. As shown in Fig. 6a, our proposed CAUSAL-MAML also adapts more quickly than PPO while demonstrating lower variance, while MAML, PRETRAINED-PPO, and $RL^2$ are affected by confounded data and fail to discover the correct path.

**Experiment 3.** In the third experiment, the agent should pick up the target color key, open the corresponding door, and reach the goal in a $18 \times 9$ grid. An illustration of the environment is provided in Fig. 5b. Key locations are uniformly generated from the set $\{(7, 2), (9, 1), (9, 4), (9, 7)\}$, door locations are uniformly generated from the set $\{(13, 1), (13, 3), (13, 5), (13, 7)\}$, and the goal are generated within the subgrid $\{(c, r) \mid 13 \leq c \leq 16, ; 6 \leq r \leq 7\}$. The wind distribution is the same as that in Experiment 1. Before obtaining the target key, approaching it yields a positive reward; after acquiring the target key, approaching the goal provides a positive reward. Fig. 6b indicates that our proposed CAUSAL-MAML outperforms PPO and MAML in terms of adaptation speed and variance reduction. MAML is able to identify the correct path, while PRETRAINED-PPO and $RL^2$ are unable to converge to the correct path.

## 5 CONCLUSION

This paper investigates a vulnerability in existing meta-reinforcement learning (meta-RL) algorithms: the challenges of unmeasured confounding in observational data. We demonstrate that when confounders are present, the standard condition of unbiased gradient estimation no longer holds, misguiding agents to learn flawed and potentially harmful policies. To address this issue, we propose a novel method for confounding-robust meta-RL. Our framework provides a principled approach to learning from confounded data by first employing causal inference techniques to reason about the possible counterfactual environments compatible with the observational data. Specifically, we train a meta-policy through direct interactions with newly generated counterfactual environments. This approach ensures that the agent learns from unbiased experiences, enabling it to acquire robust and generalizable skills. Additionally, we provide a theoretical analysis that guarantees the convergence of our algorithm. Future research could explore extending this framework to continuous action spaces and more complex, high-dimensional environments.

## REPRODUCIBILITY STATEMENT

The complete proof of all theoretical results presented in this paper, including Corol. 1 and Thm. 1, is provided in Appendix B. Detailed descriptions of the experimental setup are included in Appendix C. Readers can find all appendices as part of the supplementary text after the "References" section. All the experiments are synthetic and do not introduce any new assets. Windy Gridworld is implemented based on the Minigrid environment (Chevalier-Boisvert et al., 2023) and the Gymnasium framework (Towers et al., 2024).

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

## A  SAMPLING DETAILS OF COUNTERFACTUA CMDPS FROM THE POSTERIOR DISTRIBUTION

As discussed in the main text, CAUSAL-MAML relies on generating alternative environments sampled from the posterior distribution $\widehat{\rho}(\mathcal{M} \mid \mathcal{D}_{\text{obs}}^i)$ to enable counterfactual reasoning. In this section, we provide additional details on how to construct and sample such virtual environments.

First, we define the behavioral policy $\pi_B$ as the expectation over the exogenous variable $U$:

$$\pi_B(s, x) = \int_u \mathbb{1}_{x = f_X(s, u)} \mathcal{P}(u) \, du. \tag{10}$$

The sampled virtual CMDP $\widehat{\mathcal{M}}_i$ inherits the state space $\mathcal{S}$, action space $\mathcal{X}$, rewards $\mathcal{Y}$, and exogenous noise $\mathcal{U}$ from the original CMDP $\mathcal{M}_i$. Exogenous distribution $\widehat{\mathcal{P}}$ is estimated from the observation data $\mathcal{D}_{\text{obs}}^i$. The transition distribution $\widehat{\mathcal{T}}_i$ and expected reward function $\widehat{R}_i$ are sampled from a posterior-informed range:

$$\widehat{\mathcal{T}}_i(s, x, s') \in [\mathcal{T}_i(s, x, s')\pi_B^i(s, x), \mathcal{T}_i(s, x, s')\pi_B^i(s, x) + \pi_B^i(s, \neg x)] \tag{11}$$

$$\widehat{R}_i(s, x) \in [R_i(s, x)\pi_B^i(s, x) + a\pi_B^i(x, \neg x), R_i(s, a)\pi_B^i(s, x) + b\pi_B^i(s, \neg x)] \tag{12}$$

where $\pi_B(s, \neg x) = 1 - \pi_B(s, x)$; the original transition distribution $\mathcal{T}$ is estimated from the observational distribution $\mathcal{T}(s, x, s') = P(S_{t+1} = s' | S_t = s, X_t = x)$; the original expected reward function is given by $R(s, x) = \mathbb{E}[Y_t \mid S_t = s, X_t = x]$.

## B  PROOF DETAILS

In this section, we provide the detailed proof of the convergence of our CAUSAL-MAML method. We begin by presenting two lemmas that serve as the foundation of the proof. We then outline the proof process for these lemmas, followed by the proof of the main theorem.

### B.1  DETAILS OF CONVERGENCE PROOF

Establishing the Lipschitz property of the meta-objective function requires information from the task-specific objective functions $J_i(\theta)$, along with their gradient $\nabla_\theta J_i(\theta)$ and Hessian matrix $\nabla_\theta^2 J_i(\theta)$. Referring to the results in (Shen et al., 2019), we state the following lemmas on the Lipschitz property of the accumulated reward function $J_i(\theta)$.

**Lemma 1.** *Define $R = \max(|a|, |b|)$. Suppose Assumptions 1 and 2 hold, we have:*

    *(i) $J_i(\theta)$ is smooth with parameters $\eta_G := \frac{RG}{(1-\gamma)^2}$; that is, for any parameter $\theta \in \mathbb{R}^d$, we have $\|\nabla_\theta J_i(\theta)\| \le \eta_G$.*

    *(ii) $\nabla_\theta J_i(\theta)$ is smooth with parameters $\eta_H := \frac{(H+1)RG^2 + RL}{(1-\gamma)^2}$; that is, for any parameter $\theta \in \mathbb{R}^d$, we have $\|\nabla_\theta^2 J_i(\theta)\| \le \eta_H$.*

    *(iii) $\nabla_\theta^2 J_i(\theta)$ is smooth with parameters $\eta_\rho := \frac{2(H+1)RGL + RK}{(1-\gamma)^2}$; that is, for any parameter $\theta_1, \theta_2 \in \mathbb{R}^d$, we have $\|\nabla_\theta^2 J_i(\theta_1) - \nabla_\theta^2 J_i(\theta_2)\| \le \eta_\rho \|\theta_1 - \theta_2\|$.*

Lemma 1 demonstrates that the Lipschitz parameters of the task-specific objective function $J_i(\theta)$, its gradient $\nabla_\theta J_i(\theta)$, and its Hessian $\nabla_\theta^2 J_i(\theta)$ are $\eta_G$, $\eta_H$, $\eta_\rho$, respectively. Based on the result in Lemma 1, we can now demonstrate the Lipschitz property of the meta-objective function. The stochastic gradient of the meta-objective function is defined as follows:

$$\widehat{\nabla}_\theta F(\theta) = \frac{1}{|\mathcal{B}|} \sum_{i \in \mathcal{B}} \left( \left( I + \alpha \widehat{\nabla}_\theta^2 J_i(\theta, \widehat{\mathcal{D}}_{\text{exp,in}}^i) \right) \widehat{\nabla}_\theta J_i \left( \theta_i, \widehat{\mathcal{D}}_{\text{exp,o}}^i \right) \right.$$

$$\left. + \widehat{J}_i \left( \theta_i, \widehat{\mathcal{D}}_{\text{exp,o}}^i \right) \sum_{\tau \in \widehat{\mathcal{D}}_{\text{exp,in}}^i} \sum_{t=0}^H \nabla_\theta \log \pi(x_t \mid s_t; \theta) \right). \tag{13}$$

Referring to the result in (Fallah et al., 2021), we state the following conclusion on Lipschitz property of meta-objective function $F(\theta)$.

**Lemma 2.** *Consider the meta-objective function defined in Eq. (6) for the case that $\alpha \in (0, \frac{1}{\eta_H}]$. Suppose Assumptions 1 and 2 are satisfied. Then, we have:*

*(i)* $\widehat{\nabla}_\theta F(\theta)$ *is bounded by parameter $L_G = \frac{2RG}{(1-\gamma)^2} + \frac{D(H+1)RG}{1-\gamma}$; that is, for any parameter $\theta$, any task subset $\mathcal{B}$, and any experimental trajectory batch $\widehat{\mathcal{D}}_{exp}^i$, we have $\|\widehat{\nabla}_\theta F(\theta)\| \leq L_G$.*

*(ii)* $\widehat{\nabla}_\theta F(\theta)$ *is smooth with parameter $L_H = 4\eta_H + \alpha\eta_G\eta_\rho + D(H+1)R(\frac{L}{1-\gamma} + \frac{2G^2}{(1-\gamma^2)})$; that is, for any parameter $\theta$, any task subset $\mathcal{B}$, and any experimental trajectory batch $\widehat{\mathcal{D}}_{exp}^i$, we have $\|\widehat{\nabla}_\theta^2 F(\theta)\| \leq L_H$.*

Lemma 2 illustrates the upper bound and the Lipschitz parameter of the stochastic gradient $\widehat{\nabla}_\theta F(\theta)$.

### B.2 PROOF OF LEMMA 1

In this section, we show the proof details of Lemma 1.

**Proof of (i):**

First, we note that

$$\left\| \sum_{t=0}^H \nabla_\theta \log \pi(x_t \mid s_t; \theta) \Psi_t \right\| \leq \sum_{t=0}^H \|\nabla_\theta \log \pi(x_t \mid s_t; \theta)\| |\Psi_t|$$

$$\leq \sum_{t=0}^H |\Psi_t| G.$$

The accumulated reward is

$$|\Psi_t| = \left| \sum_{t'=t}^H \gamma^t R_i(s_{t'}, x_{t'}) \right|$$

$$\leq R \sum_{t'=t}^H \gamma^{t'}$$

$$\leq \frac{R\gamma^{t'}}{1-\gamma}.$$

Consequently, we have

$$\left\| \sum_{t=0}^H \nabla_\theta \log \pi(x_t \mid s_t; \theta) \Psi_t \right\| \leq RG \sum_{t=0}^H \frac{\gamma^{t'}}{1-\gamma}$$

$$\leq \frac{RG}{(1-\gamma)^2}.$$

**Proof of (ii):**

Note that

$$\left\|(\sum_{t=0}^{H}\nabla_\theta\log\pi(x_t\mid s_t;\theta)\Psi_t)\nabla_\theta\log q_i(\tau;\theta)^\mathsf{T}+\sum_{t=0}^{H}\nabla_\theta^2\log\pi(x_t\mid s_t;\theta)\Psi_t\right\|$$

$$\leq\left\|\sum_{t=0}^{H}\nabla_\theta\log\pi(x_t\mid s_t;\theta)\Psi_t)\right\|\|\nabla_\theta\log q_i(\tau;\theta)\right\|+\left\|\sum_{t=0}^{H}\nabla_\theta^2\log\pi(x_t\mid s_t;\theta)\Psi_t\right\|.$$

First, we consider the bound on $\|\nabla_\theta\log q_i(\tau;\theta)\|$:

$$\|\nabla_\theta\log q_i(\tau;\theta)\|=\sum_{t=0}^{H}\|\nabla_\theta\log\pi(x_t\mid s_t;\theta)\|$$

$$\leq(H+1)G$$

According to the result in Lemma 1(i),

$$\left\|\sum_{t=0}^{H}\nabla_\theta\log\pi(x_t\mid s_t;\theta)\Psi_t\right\|\leq\frac{RG}{(1-\gamma)^2}.$$

Then, we consider the bound on $\|\sum_{t=0}^{H}\nabla_\theta^2\log\pi(x_t\mid s_t;\theta)\Psi_t\|$:

$$\left\|\sum_{t=0}^{H}\nabla_\theta^2\log\pi(x_t\mid s_t;\theta)\Psi_t\right\|\leq\sum_{t=0}^{H}\|\nabla_\theta^2\log\pi(x_t\mid s_t;\theta)\|\|\Psi_t|$$

$$\leq RL\sum_{t=0}^{H}\frac{\gamma^{t'}}{1-\gamma}$$

$$\leq\frac{LR}{(1-\gamma)^2}.$$

Consequently, we have

$$\left\|(\sum_{t=0}^{H}\nabla_\theta\log\pi(x_t\mid s_t;\theta)\Psi_t)\nabla_\theta\log q_i(\tau;\theta)^\mathsf{T}+\sum_{t=0}^{H}\nabla_\theta^2\log\pi(x_t\mid s_t;\theta)\Psi_t\right\|$$

$$\leq\frac{(H+1)RG^2+RL}{(1-\gamma)^2}.$$

**Proof of (iii):** Note that

$$\|\nabla_\theta^2 J_i(\theta_1)-\nabla_\theta^2 J_i(\theta_2)\|$$

$$\leq\left\|\sum_{t=0}^{H}\nabla_\theta\log\pi(x_t\mid s_t;\theta_1)\Psi_t\nabla_\theta\log q_i(\tau;\theta_1)^\mathsf{T}-\sum_{t=0}^{H}\nabla_\theta\log\pi(x_t\mid s_t;\theta_2)\Psi_t\nabla_\theta\log q_i(\tau;\theta_2)^\mathsf{T}\right\|$$

$$+\left\|\sum_{t=0}^{H}\nabla_\theta^2\log\pi(x_t\mid s_t;\theta_1)\Psi_t-\sum_{t=0}^{H}\nabla_\theta^2\log\pi(x_t\mid s_t;\theta_2)\Psi_t\right\|$$

$$\leq\|\nabla_\theta\log q_i(\tau;\theta)\|\left\|\sum_{t=0}^{H}\nabla_\theta\log\pi(x_t\mid s_t;\theta_1)\Psi_t-\sum_{t=0}^{H}\nabla_\theta\log\pi(x_t\mid s_t;\theta_2)\Psi_t\right\|$$

$$+\left\|\sum_{t=0}^{H}\nabla_\theta\log\pi(x_t\mid s_t;\theta_1)\Psi_t\right\|\|\nabla_\theta\log q_i(\tau;\theta_1)-\nabla_\theta\log q_i(\tau;\theta_2)\right\|$$

$$+\left\|\sum_{t=0}^{H}\nabla_\theta^2\log\pi(x_t\mid s_t;\theta_1)\Psi_t-\sum_{t=0}^{H}\nabla_\theta^2\log\pi(x_t\mid s_t;\theta_2)\Psi_t\right\|.$$

First, we consider the Lipschitz parameter of $\sum_{t=0}^{H} \nabla_\theta \log \pi(x_t \mid s_t; \theta) \Psi_t$:

$$\left\| \sum_{t=0}^{H} \nabla_\theta \log \pi(x_t \mid s_t; \theta_1) \Psi_t - \sum_{t=0}^{H} \nabla_\theta \log \pi(x_t \mid s_t; \theta_2) \Psi_t \right\|$$

$$\leq \sum_{t=0}^{H} \| \nabla_\theta \log \pi(x_t \mid s_t; \theta_1) - \nabla_\theta \log \pi(x_t \mid s_t; \theta_2) \| |\Psi_t|.$$

According to Assumption 1, the gradient of logarithmic policy is smooth with parameter $L$, i.e.,

$$\| \nabla_\theta \log \pi(x_t \mid s_t; \theta_1) - \nabla_\theta \log \pi(x_t \mid s_t; \theta_2) \| \leq L \|\theta_1 - \theta_2\|.$$

Therefore,

$$\left\| \sum_{t=0}^{H} \nabla_\theta \log \pi(x_t \mid s_t; \theta_1) \Psi_t - \sum_{t=0}^{H} \nabla_\theta \log \pi(x_t \mid s_t; \theta_2) \Psi_t \right\|$$

$$\leq L \|\theta_1 - \theta_2\| \sum_{t=0}^{H} \frac{R \gamma^{t'}}{1 - \gamma}$$

$$\leq \frac{RL}{(1 - \gamma)^2} \|\theta_1 - \theta_2\|.$$

It is obvious that $\nabla_\theta \log q_i(\tau; \theta)$ is Lipschitz with parameter $(H + 1)L$, i.e.,

$$\| \nabla_\theta \log q_i(\tau; \theta_1) - \nabla_\theta \log q_i(\tau; \theta_2) \| \leq (H + 1)L \|\theta_1 - \theta_2\|.$$

According to Assumption 2, wherein the gradient of the logarithmic policy is smooth with parameter $K$, we have a similar conclusion as in the above proof:

$$\left\| \sum_{t=0}^{H} \nabla_\theta^2 \log \pi(x_t \mid s_t; \theta_1) \Psi_t - \sum_{t=0}^{H} \nabla_\theta^2 \log \pi(x_t \mid s_t; \theta_2) \Psi_t \right\| \leq \frac{RK}{(1 - \gamma)^2} \|\theta_1 - \theta_2\|.$$

From the proof of Lemma 1(ii), we know the bound $\| \nabla_\theta \log q_i(\tau; \theta) \| \leq (H + 1)G$. The result in Lemma 1(i) shows that $\| \sum_{t=0}^{H} \nabla_\theta \log \pi(x_t \mid s_t; \theta) \Psi_t \| \leq \frac{RG}{(1 - \gamma)^2}$. Finally, these yield the result that

$$\| \nabla_\theta^2 J_i(\theta_1) - \nabla_\theta^2 J_i(\theta_2) \| \leq \left( (H + 1)G \frac{RL}{(1 - \gamma)^2} + \frac{RG}{(1 - \gamma)^2} (H + 1)L + \frac{RK}{(1 - \gamma)^2} \right) \|\theta_1 - \theta_2\|$$

$$= \frac{2(H + 1)RGL + RK}{(1 - \gamma)^2} \|\theta_1 - \theta_2\|.$$

### B.3 PROOF OF LEMMA 2

In this section, we show the proof details of Lemma 2.

**Proof of (i):** We first note that

$$\| \nabla_\theta F(\theta) \| = \| (I + \alpha \widehat{\nabla}_\theta^2 J_i(\theta, \widehat{\mathcal{D}}_{\exp}^i)) \nabla_\theta J_i(\theta + \alpha \widehat{\nabla} J_i(\theta, \widehat{\mathcal{D}}_{\exp}^i))$$

$$+ J_i(\theta + \alpha \widehat{\nabla} J_i(\theta, \widehat{\mathcal{D}}_{\exp}^i)) \sum_{\tau \in \widehat{\mathcal{D}}_{\exp}^i} \sum_{t=0}^{H} \nabla_\theta \log \pi_\theta(x_t \mid s_t; \theta) \|$$

$$\leq \| I + \alpha \widehat{\nabla}_\theta^2 J_i(\theta, \widehat{\mathcal{D}}_{\exp}^i) \| \| \nabla_\theta J_i(\theta + \alpha \widehat{\nabla} J_i(\theta, \widehat{\mathcal{D}}_{\exp}^i)) \|$$

$$+ \| J_i(\theta + \alpha \widehat{\nabla} J_i(\theta, \widehat{\mathcal{D}}_{\exp}^i)) \| \left\| \sum_{\tau \in \widehat{\mathcal{D}}_{\exp}^i} \sum_{t=0}^{H} \nabla_\theta \log \pi_\theta(x_t \mid s_t; \theta) \right\|.$$

Lemma 1 implies that $\|\nabla_\theta J_i(\theta - \alpha\widehat{\nabla} J_i(\theta, \widehat{\mathcal{D}}^i_{\exp}))\| \leq \eta_G$. For any parameter $\theta$, the accumulated reward function is bounded by

$$\|J_i(\theta)\| = \|\sum_{t=0}^{H} \gamma^t R_i(s_t, x_t)]\|$$

$$\leq R \sum_{t=0}^{H} \gamma^t$$

$$\leq \frac{R}{1-\gamma}.$$

Recalling Assumption 1, we have that $\|\sum_{\tau \in \widehat{\mathcal{D}}^i_{\exp}} \sum_{t=0}^{H} \nabla_\theta \log \pi_\theta(s_t, x_t; \theta)\|$ is bounded by $GD(H+1)$. $(I + \alpha\widehat{\nabla}^2_\theta J_i(\theta, \widehat{\mathcal{D}}^i_{\exp}))$ is bounded by $1 + \alpha\eta_H$. Relying on the assumption $\alpha \leq \eta_H$, we know $(1 + \alpha\eta_H) \leq 2$. Now, we know that the gradient of the objective function $\|\nabla_\theta F(\theta)\|$ is bounded by $2\eta_G + \frac{(H+1)DRG}{1-\gamma} = \frac{2RG}{(1-\gamma)^2} + \frac{D(H+1)RG}{1-\gamma}$.

**Proof of (ii):**

The Lipschitz parameter of $\widehat{\nabla}_\theta F(\theta)$ is the sum of the Lipschitz parameters of $(I + \alpha\widehat{\nabla}^2_\theta J_i(\theta, \widehat{\mathcal{D}}^i_{\exp}))\nabla_\theta J_i(\theta + \alpha\widehat{\nabla}_\theta J_i(\theta, \widehat{\mathcal{D}}^i_{\exp}))$ and $J_i(\theta + \alpha\widehat{\nabla} J_i(\theta, \widehat{\mathcal{D}}^i_{\exp})) \sum_{\tau \in \widehat{\mathcal{D}}^i_{\exp}} \sum_{t=0}^{H} \nabla_\theta \log \pi(x_t \mid s_t; \theta)$. Next, we analyze each item separately.

Consider the Lipschitz parameter of $(I + \alpha\widehat{\nabla}^2_\theta J_i(\theta, \widehat{\mathcal{D}}^i_{\exp}))\nabla_\theta J_i(\theta + \alpha\widehat{\nabla} J_i(\theta, \widehat{\mathcal{D}}^i_{\exp}))$. We have

$$\|(I + \alpha\widehat{\nabla}^2_\theta J_i(\theta_1, \widehat{\mathcal{D}}^i_{\exp}))\nabla_\theta J_i(\theta_1 + \alpha\widehat{\nabla} J_i(\theta_1, \widehat{\mathcal{D}}^i_{\exp}))$$
$$- (I + \alpha\widehat{\nabla}^2_\theta J_i(\theta_2, \widehat{\mathcal{D}}^i_{\exp}))\nabla_\theta J_i(\theta_2 + \alpha\widehat{\nabla} J_i(\theta_2, \widehat{\mathcal{D}}^i_{\exp}))\|$$
$$\leq \|(I + \alpha\widehat{\nabla}^2_\theta J_i(\theta, \widehat{\mathcal{D}}^i_{\exp}))\|\|\nabla_\theta J_i(\theta_1 + \alpha\widehat{\nabla} J_i(\theta_1, \widehat{\mathcal{D}}^i_{\exp})) - \nabla_\theta J_i(\theta_2 + \alpha\widehat{\nabla} J_i(\theta_2, \widehat{\mathcal{D}}^i_{\exp}))\|$$
$$+ \|\nabla_\theta J_i(\theta + \alpha\widehat{\nabla} J_i(\theta, \widehat{\mathcal{D}}^i_{\exp}))\|\|\alpha\widehat{\nabla}^2_\theta J_i(\theta_1, \widehat{\mathcal{D}}^i_{\exp}) - \alpha\widehat{\nabla}^2_\theta J_i(\theta_2, \widehat{\mathcal{D}}^i_{\exp})\|.$$

According to the result in Lemma 1, we know that $(I + \alpha\widehat{\nabla}^2_\theta J_i(\theta, \widehat{\mathcal{D}}^i_{\exp}))$ is bounded by $(1 + \alpha\eta_H)$ and smooth with parameter $\alpha\eta_\rho$. $\nabla_\theta J_i(\theta)$ is bounded by $\eta_G$ and smooth with parameter $\eta_H$. Along with the fact that the Lipschitz parameter of the combination of functions is the product of their Lipschitz parameters and $\theta + \alpha\widehat{\nabla} J_i(\theta, \widehat{\mathcal{D}}^i_{\exp})$ is smooth with parameter $1 + \alpha\eta_H$, $\nabla_\theta J_i(\theta + \alpha\widehat{\nabla} J_i(\theta, \widehat{\mathcal{D}}^i_{\exp}))$ is smooth with parameter $(1 + \alpha\eta_H)\eta_H$. Therefore,

$$\|(I + \alpha\widehat{\nabla}^2_\theta J_i(\theta_1, \widehat{\mathcal{D}}^i_{\exp}))\nabla_\theta J_i(\theta_1 + \alpha\widehat{\nabla} J_i(\theta_1, \widehat{\mathcal{D}}^i_{\exp}))$$
$$- (I + \alpha\widehat{\nabla}^2_\theta J_i(\theta_2, \widehat{\mathcal{D}}^i_{\exp}))\nabla_\theta J_i(\theta_2 + \alpha\widehat{\nabla} J_i(\theta_2, \widehat{\mathcal{D}}^i_{\exp}))\|$$
$$\leq (1 + \alpha\eta_H)(1 + \alpha\eta_H)\eta_H\|\theta_1 - \theta_2\| + \eta_G(\alpha\eta_\rho)\|\theta_1 - \theta_2\|$$
$$= ((1 + \alpha\eta_H)^2\eta_H + \alpha\eta_G\eta_\rho)\|\theta_1 - \theta_2\|.$$

Using the assumption $\alpha \leq \eta_H$, we know $(1 + \alpha\eta_H) \leq 2$. Consequently, $(I + \alpha\widehat{\nabla}^2_\theta J_i(\theta, \widehat{\mathcal{D}}^i_{\exp}))\nabla_\theta J_i(\theta + \alpha\widehat{\nabla}_\theta J_i(\theta, \widehat{\mathcal{D}}^i_{\exp}))$ is smooth with parameter $4\eta_H + \alpha\eta_G\eta_\rho$.

Now consider the Lipschitz parameter of $J_i(\theta + \alpha\widehat{\nabla}J_i(\theta, \widehat{\mathcal{D}}_{\exp}^i)) \sum_{\tau \in \widehat{\mathcal{D}}_{\exp}^i} \sum_{t=0}^H \nabla_\theta \log \pi(x_t \mid s_t; \theta)$:

$$\|J_i(\theta_1 + \alpha\widehat{\nabla}J_i(\theta_1, \widehat{\mathcal{D}}_{\exp}^i)) \sum_{\tau \in \widehat{\mathcal{D}}_{\exp}^i} \sum_{t=0}^H \nabla_\theta \log \pi(x_t \mid s_t; \theta_1)$$

$$- J_i(\theta_2 + \alpha\widehat{\nabla}J_i(\theta_2, \widehat{\mathcal{D}}_{\exp}^i)) \sum_{\tau \in \widehat{\mathcal{D}}_{\exp}^i} \sum_{t=0}^H \nabla_\theta \log \pi(x_t \mid s_t; \theta_2)\|$$

$$\leq \|J_i(\theta + \alpha\widehat{\nabla}J_i(\theta, \widehat{\mathcal{D}}_{\exp}^i))\| \sum_{\tau \in \widehat{\mathcal{D}}_{\exp}^i} \sum_{t=0}^H \|\nabla_\theta \log \pi(x_t \mid s_t; \theta_1) - \nabla_\theta \log \pi(x_t \mid s_t; \theta_2)\|$$

$$+ \sum_{\tau \in \widehat{\mathcal{D}}_{\exp}^i} \sum_{t=0}^H \|\nabla_\theta \log \pi(x_t \mid s_t; \theta)\| \|J_i(\theta_1 + \alpha\widehat{\nabla}J_i(\theta_1, \widehat{\mathcal{D}}_{\exp}^i)) - J_i(\theta_2 + \alpha\widehat{\nabla}J_i(\theta_2, \widehat{\mathcal{D}}_{\exp}^i))\|.$$

Relying on the Assumption 2, we know that $\nabla_\theta \log \pi(x_t \mid s_t; \theta)$ is bounded by $G$ and smooth with parameter $L$. Along with the fact that the Lipschitz parameter of the combination of functions is the product of their Lipschitz parameters and $\theta + \alpha\widehat{\nabla}J_i(\theta, \widehat{\mathcal{D}}_{\exp}^i)$ is smooth with parameter $1 + \alpha\eta_H$, $J_i(\theta + \alpha\widehat{\nabla}J_i(\theta, \widehat{\mathcal{D}}_{\exp}^i))$ is smooth with parameter $(1 + \alpha\eta_H)\eta_G \leq 2\eta_G$. Therefore,

$$\|J_i(\theta_1 + \alpha\widehat{\nabla}J_i(\theta_1, \widehat{\mathcal{D}}_{\exp}^i)) \sum_{\tau \in \widehat{\mathcal{D}}_{\exp}^i} \sum_{t=0}^H \nabla_\theta \log \pi(x_t \mid s_t; \theta_1)$$

$$- J_i(\theta_2 + \alpha\widehat{\nabla}J_i(\theta_2, \widehat{\mathcal{D}}_{\exp}^i)) \sum_{\tau \in \widehat{\mathcal{D}}_{\exp}^i} \sum_{t=0}^H \nabla_\theta \log \pi(x_t \mid s_t; \theta_2)\|$$

$$\leq \frac{R}{1-\gamma} D(H+1)L\|\theta_1 - \theta_2\| + D(H+1)G2\eta_G\|\theta_1 - \theta_2\|$$

$$= D(H+1)R\left(\frac{L}{1-\gamma} + \frac{2G^2}{(1-\gamma^2)}\right).$$

According to the following derivation, we know that the Lipschitz parameter of $J_i(\theta + \alpha\widehat{\nabla}J_i(\theta, \widehat{\mathcal{D}}_{\exp}^i)) \sum_{\tau \in \widehat{\mathcal{D}}_{\exp}^i} \sum_{t=0}^H \nabla_\theta \log \pi(x_t \mid s_t; \theta)$ is $D(H+1)R\left(\frac{L}{1-\gamma} + \frac{2G^2}{(1-\gamma^2)}\right)$.

Finally, the Lipschitz parameter of $\nabla_\theta F(\theta)$ is $4\eta_H + \alpha\eta_G\eta_\rho + D(H+1)R(\frac{L}{1-\gamma} + \frac{2G^2}{(1-\gamma^2)})$.

### B.4 Proof of Theorem 1

First, we establish an upper bound on the variance of the estimation of the meta-objective function gradient $\nabla_\theta F(\theta)$.

**Lemma 3.** *Suppose that the conditions in Assumptions.1, 2 are satisfied. For the case that $\alpha \in (0, \frac{1}{\eta_H}]$, and any choice of task subset $\mathcal{B}$, we have*

$$\mathbb{E}\|\widehat{\nabla}_\theta F(\theta) - \nabla_\theta F(\theta)\| \leq \frac{L_G^2}{BD}.$$

The proof is based on an application of the law of large numbers and variance additivity. If $\{X_1, X_2, \ldots, X_n\}$ are independent random variables with $\mathbb{E}[X_i] = \mu$, and variance bounded by $\text{Var}[X_i] \leq \sigma^2$, then the variance of the sample mean is bounded by

$$\mathbb{E}\left[\left\|\frac{X_1 + \cdots + X_n}{n} - \mu\right\| \leq \frac{\sigma^2}{n}\right].$$

Next, we proceed with the proof. Using the smoothness property of $\nabla_\theta F(\theta)$, we have

$$|F(\theta_{k+1}) - F(\theta_k) - \nabla_\theta F(\theta_k) \times (\theta_{k+1} - \theta_k)| \le \frac{L_H^2}{2}\|\theta_{k+1} - \theta_k\|.$$

At iteration $k + 1$, we have $\theta_{k+1} - \theta_k = \beta\widehat{\nabla}_\theta F(\theta_k)$, and therefore,

$$-F(\theta_{k+1}) \le -F(\theta_k) - \beta\nabla_\theta F(\theta_k) \times \widehat{\nabla}_\theta F(\theta_k) + \frac{L_H^2}{2}\beta^2\|\widehat{\nabla}_\theta F(\theta_k)\|^2.$$

Taking the expectations of both sides, we obtain

$$-\mathbb{E}[F(\theta_{k+1})] \le -\mathbb{E}[F(\theta_k)] - \beta\mathbb{E}[\|\nabla_\theta F(\theta_k)\|^2]$$
$$+ \frac{L_H^2}{2}\beta^2(\mathbb{E}[\|\nabla_\theta F(\theta_k)\|^2] + \mathbb{E}[\|\widehat{\nabla}_\theta F(\theta) - \nabla_\theta F(\theta)\|^2])$$
$$\le -\mathbb{E}[F(\theta_k)] - \frac{\beta}{2}\mathbb{E}[\|\nabla_\theta F(\theta_k)\|^2] + \frac{L_G^2 L_H \beta^2}{2BD}.$$

We prove the conclusion by contradiction. Assume our result does not hold for the first $T$ iterations, i.e.,

$$\mathbb{E}[\|\nabla_\theta F(\theta_k)\|^2] \ge \frac{2L_G^2 L_H \beta}{BD} + \epsilon^2.$$

For any $0 \le k \le T - 1$, we have

$$-\mathbb{E}[F(\theta_{k+1})] \le -\mathbb{E}[F(\theta_k)] - \frac{\beta\epsilon^2}{2} - \frac{L_G^2 L_H \beta^2}{BD} + \frac{L_G^2 L_H \beta^2}{2BD}.$$

Summing up the above formulation for $k = 0, \ldots, T - 1$, we obtain

$$-\mathbb{E}[F(\theta_T)] \le -\mathbb{E}[F(\theta_0)] - T(\frac{\beta\epsilon^2}{2} + \frac{L_G^2 L_H \beta^2}{2BD}).$$

We know that $\mathbb{E}[F(\theta)] \in [\frac{a}{1-\gamma}, \frac{b}{1-\gamma}]$, and hence $\mathbb{E}[F(\theta_0)] - \mathbb{E}[F(\theta_T)] \le \frac{b-a}{1-\gamma}$. Then, we have

$$T\left(\frac{\beta\epsilon^2}{2} + \frac{L_G^2 L_H \beta^2}{2BD}\right) \le \frac{b-a}{1-\gamma}.$$

When we choose $T \ge \frac{b-a}{1-\gamma}(\frac{2}{\beta\epsilon^2} + \frac{2BD}{L_G^2 L_H \beta^2})$, contradiction occurs. Hence, the desired result follows.

## C    EXPERIMENTAL DETAILS

In this section, we provide the details of the baseline methods. We also introduce a new baseline for comparison with our method in the same environments and show the corresponding results.

### C.1    BASELINES

The baseline algorithms, Standard MAML and Pre-trained PPO, are presented in Algorithms 2 and 3, respectively. The new baseline, Causal PPO, is introduced in Algorithm 4.

---

**Algorithm 2:** MAML

1 **Require:** Initial parameter $\theta$
2 **while** not done **do**
3      Nature samples a batch of CMDP tasks $\mathcal{B} = \{\mathcal{M}_i\}_{i=1}^B$ from distribution $\rho(\mathcal{M})$
4      **for** all task $\mathcal{M}_i \in \mathcal{B}$ **do**
5          Sample observation trajectories $\mathcal{D}_{obs,in}^i$ and $\mathcal{D}_{obs,o}^i$ in environment $\mathcal{M}_i$
6          Compute inner gradient $\widehat{\nabla}_\theta J_i(\theta, \mathcal{D}_{in}^i)$ using dataset $\mathcal{D}_{in}^i$
7          Set adapted parameter $\theta_i = \theta + \alpha\widehat{\nabla}_\theta J_i(\theta, \mathcal{D}_{in}^i)$
8      **end**
9      Update $\theta \leftarrow \theta + \beta\widehat{\nabla}_\theta \sum_{i=1}^B J_i(\theta_i, \mathcal{D}_o^i)$
10 **end**

---

---

**Algorithm 3:** PRE-TRAINED PPO

**1 Require:** Initial parameter $\theta$
**2 while** not done **do**
**3**      Nature samples a batch of CMDP tasks $\mathcal{B} = \{\mathcal{M}_i\}_{i=1}^B$ from distribution $\rho(\mathcal{M})$
**4**      **for** all task $\mathcal{C}_i \in \mathcal{B}$ **do**
**5**          Sample observation trajectories $\mathcal{D}^i$ in environment $\mathcal{M}_i$
**6**          Compute gradient $\widehat{\nabla}_\theta J_i(\theta, \mathcal{D}^i)$ using dataset $\mathcal{D}^i$
**7**          Update parameter $\theta \leftarrow \theta + \alpha \widehat{\nabla} J_i(\theta, \mathcal{D}^i)$
**8**      **end**
**9 end**

---

**Algorithm 4:** Causal PPO

**1 Require:** Initial parameter $\theta$, an approximate prior over CMDPs $\widehat{\rho}(\mathcal{M})$
**2 while** not done **do**
**3**      Nature samples a batch of CMDP tasks $\mathcal{B} = \{\mathcal{M}_i\}_{i=1}^B$ from distribution $\rho(\mathcal{M})$
**4**      **for** all task $\mathcal{C}_i \in \mathcal{B}$ **do**
**5**          Sample a new environment $\widehat{\mathcal{M}}_i$ from the posterior $\widehat{\rho}(\mathcal{M} \mid \mathcal{D}^i_{\text{obs}})$
**6**          Sample experimental trajectories $\widehat{\mathcal{D}}^i_{\text{exp}}$ using agent policy $\pi(\cdot \mid \cdot; \theta)$ in environment $\widehat{\mathcal{M}}_i$
**7**          Compute gradient $\widehat{\nabla}_\theta J_i(\theta, \widehat{\mathcal{D}}^i_{\text{exp}})$ using dataset $\widehat{\mathcal{D}}^i_{\text{exp}}$
**8**          Update parameter $\theta \leftarrow \theta + \alpha \widehat{\nabla} J_i(\theta, \widehat{\mathcal{D}}^i_{\text{exp}})$
**9**      **end**
**10 end**

---

### C.2    COMPARISON OF CAUSAL-MAML AND CAUSAL PRE-TRAINED PPO

We also compare the causal PPO method with our causal MAML method. Causal PPO also constructs virtual environments using demonstrator data in confounding MDPs. Then causal PPO collects experimental data using policy $\pi_\theta$ in such virtual environments and update parameters by gradients calculated on these experimental trajectories. Fig.7a and Fig.7c show that causal PPO have almost the same performance as our proposed causal MAML, including the similar adaption speed and variance. Fig.7b indicates that causal PPO adapts more quickly than our proposed causal MAML, however, with a larger variance in returns during adaption. (Zhao et al., 2022) and (Gao & Sener, 2020) reveal the same results: multi-task pretraining performs equally, or even better than meta-pretraining for adapting to new tasks.

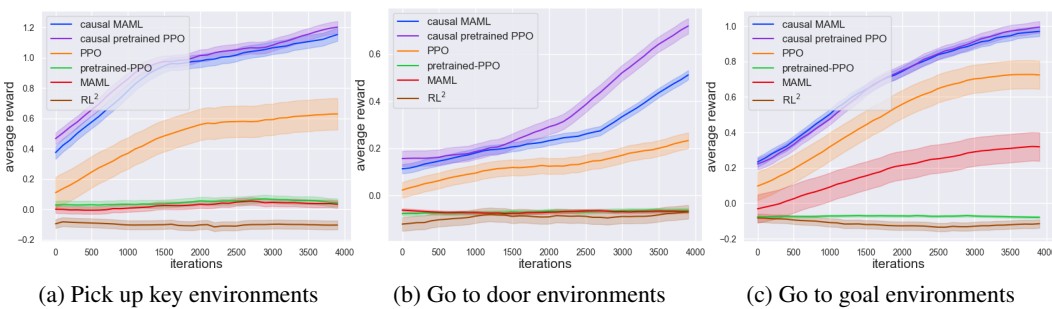

(a) Pick up key environments      (b) Go to door environments      (c) Go to goal environments

Figure 7: Returns of MiniGrid environments comparing PPO from scratch, Pre-Trained PPO, standard MAML, CAUSAL PRE-TRAINED PPO, and Proposed Causal-MAML with error bars

Table 1 further summarizes the average testing returns across three environments. The results show that both Causal PPO and Causal MAML significantly outperform standard baselines such as PRE-TRAINED PPO, Standard MAML, and PPO from scratch. Notably, Causal-PPO achieves the highest

Table 1: Average testing returns of CAUSAL-MAML against baselines.

| METHOD | Pick-Up-Key | Go-To-Door | Go-To-Goal |
|---|---|---|---|
| $RL^2$ | -0.10±0.12 | -0.09±0.12 | -0.12±0.01 |
| PRE-TRAINED PPO | 0.05±0.10 | -0.06±0.03 | -0.07±0.02 |
| Stardard MAML | 0.02±0.10 | -0.07±0.03 | 0.19±0.32 |
| PPO from scratch | 0.65±0.41 | 0.26±0.13 | 0.69±0.32 |
| CAUSAL PRE-TRAINED PPO | **1.28±0.16** | **0.82±0.13** | **1.05±0.13** |
| CAUSAL-MAML | 1.21±0.17 | 0.65±0.08 | 1.00±0.11 |

returns overall, while Causal MAML attains competitive performance with slightly lower variance in certain tasks.

We also compare our causal meta-RL methods with $RL^2$(Duan et al., 2017), a well-known meta-RL baseline. Fig.7a, Fig.7b, Fig.7c show that $RL^2$ fails to learn a useful policy in confounding enviroments. Table 1 further summarizes the average testing returns and standard deviation of $RL^2$. Regarding the performance of $RL^2$, we believe the key issue is that recurrent policies depend solely on observation trajectories generated by a behavior policy interacting with candidate environments. Due to the presence of unobserved confounders, these observations may include spurious correlations between actions and subsequent outcomes, which hinders accurate estimation of actual causal effects, such as state-action values.

