# OpenReview forum: "Confounding Robust Meta-Reinforcement Learning: A Causal Approach"
_ICLR.cc/2026/Conference — Submitted to ICLR 2026_

### Official Review · Reviewer_e8oJ · 2025-10-28

**Soundness:** 3
**Presentation:** 2
**Contribution:** 2
**Rating:** 4
**Confidence:** 2

**Summary:**

This paper tackles unobserved confounders in Meta-RL via partial identification methods to generate counterfactual trajectories from candidate environments that align with the confounded observations.

**Strengths:**

This paper addresses an important challenge in reinforcement learning: the presence of unobserved confounders, approached from a causal inference perspective. Numerical experiments are conducted to demonstrate the effectiveness of the proposed method.

**Weaknesses:**

1. The confounding environment appears to be overly simplified. In many sequential settings, the transition dynamics are modeled as a function  $f: \mathcal{S} \times \mathcal{X} \times \mathcal{U} \rightarrow \mathcal{S} \times \mathcal{U}$, whereas the paper assumes that the current unobserved state is not influenced by the history and it somehow does not reflect the challenge in the sequential setting. This assumption restricts the applicability of the proposed method. I am wondering whether the methods extend to this more general setting, and if so, what additional assumptions or modifications (e.g., on the evolution of $\mathcal{U}$, identifiability, or estimation strategy) are required.

2.  There appear to be important missing components in the paper. In particular, no formal identification results are established for the counterfactual trajectories. It remains unclear what additional assumptions and regularity conditions are required to ensure identification, and what the theoretical guarantees are regarding the quality of the estimated counterfactual trajectories. Furthermore, in practical applications, it is not evident how to determine the dimension of the unobserved states.

3. The paper claims that the solution minimizes the generalization error. However, the theoretical results only show for the first-order stationary point. More argument and discussion are needed for this statement.

**Questions:**

See above.

**Details Of Ethics Concerns:**

N/A.

---

> ### Author Response · Authors · 2025-11-20
>
> We thank the reviewer for the feedback and appreciate your recognition of the significance of our work. We have addressed each of your questions in the responses below.
>
>
> > **W1** “The confounding environment appears to be overly simplified. In many sequential settings, the transition dynamics are modeled as a function , whereas the paper assumes that the current unobserved state is not influenced by the history and it somehow does not reflect the challenge in the sequential setting. This assumption restricts the applicability of the proposed method. I am wondering whether the methods extend to this more general setting, and if so, what additional assumptions or modifications (e.g., on the evolution of , identifiability, or estimation strategy) are required.”
>
> This paper focuses on the model of confounded MDPs, which is a significant generalization of the standard MDP framework that allows for the presence of unobserved confounding factors. Given that MDPs are widely used across various applications, we anticipate that the proposed method will be applicable to many real-life scenarios. Additionally, various experiments conducted in unstructured image domains show that policies learned in confounded MDPs are robust against model misspecification of the Markov property and remain effective in confounded POMDP environments.
>
> Finally, we acknowledge that meta-RL in confounded non-Markovian environments presents a significant challenge. Similar to current strategies for extending Markov policies to POMDPs (as discussed by Hausknecht & Stone, 2017), one potential approach for initial exploration could involve replacing convolutional neural network (CNN) policy networks with recurrent architectures such as Long Short-Term Memory (LSTM) networks, while keeping the rest of the algorithms unchanged.
>
> > **W2** “There appear to be important missing components in the paper. In particular, no formal identification results are established for the counterfactual trajectories. It remains unclear what additional assumptions and regularity conditions are required to ensure identification, and what the theoretical guarantees are regarding the quality of the estimated counterfactual trajectories. Furthermore, in practical applications, it is not evident how to determine the dimension of the unobserved states.”
>
> First, we would like to clarify that our proposed method employs partial identification techniques (Manski, 1989), which do not require any additional identification assumptions. We begin by constructing informative intervals from confounded observations that encompass the true dynamics of the system. Next, we incorporate these bounds into meta-RL algorithms to facilitate the learning of a policy initialization that generalizes effectively across diverse environments using only a few experimental samples.
>
> The theoretical guarantees concerning the robustness of these bounds are outlined in Corollary 1, while Theorem 1 provides the convergence results for the meta-RL policy initialization. Additionally, Definition 2 specifies a sufficient upper bound on the latent state cardinality, which is expressed as a polynomial function of the number of observed states.
>
> In practice, when both the observed and latent states are high-dimensional, we present analytical solutions in Appendix A to help approximate the causal bounds. In this scenario, the posterior sampling in Causal-MAML can be done with uniform sampling from these closed-form bounds.
>
> > **W3** “The paper claims that the solution minimizes the generalization error. However, the theoretical results only show for the first-order stationary point. More argument and discussion are needed for this statement.”
>
> Similar to the existing theoretical framework (Fallah et al., 2019), our analysis follows the common practice of establishing convergence to a first-order stationary point, which is standard in non-convex optimization with neural networks. The network architecture and activation functions used in our method satisfy the required smoothness assumptions (e.g., second-order Lipschitz continuity).

---

### Official Review · Reviewer_S7t1 · 2025-10-29

**Soundness:** 3
**Presentation:** 3
**Contribution:** 2
**Rating:** 4
**Confidence:** 4

**Summary:**

This work aims to address the problem of unobserved confounders in meta-reinforcement learning environments. By leveraging the method of partial counterfactual identification, the authors propose a causal MAML framework, which utilizes counterfactual trajectories to find a policy initialization that exhibits strong generalization performance in target domains.

**Strengths:**

1.	The paper addresses an important issue in meta-reinforcement learning, confounding bias, and introduces a causal perspective to tackle it.
2.	The proposed algorithm is rigorously derived through formal theoretical development, including the definitions of CMDPs and canonical causal models, as well as a convergence proof under bounded-gradient assumptions.

**Weaknesses:**

1.	The entire study is conducted under discrete and finite CMDP settings. Consequently, both theoretical formulation and empirical validation are confined to simplified, low-dimensional environments. While the framework is theoretically sound, its applicability and scalability to high-dimensional continuous control tasks remain unverified.
2.	The literature review could be more comprehensive. While the paper states that research on handling unobserved confounders in meta-reinforcement learning is still missing, several existing works have already investigated this direction using causal approaches.

**Questions:**

1.	Not disclosing significant LLM usage.
2.	In lines 086–090, the paper identifies the lack of “a systematic approach for performing meta-learning across heterogeneous domains with the presence of unmeasured confounding.”
However, several recent studies [1,2] have already explored causal approaches to address unobserved confounders in meta-reinforcement learning.
Therefore, the gap and motivation would be stronger if the authors explicitly acknowledge these prior works and clarify how their method fundamentally differs from, or advances beyond, existing causal meta-RL approaches.

1.	‘The shorter purple route’ should be ‘The … orange …’ in line 209.
2.	The paper does not include comparisons with existing meta reinforcement learning based on casual methods.

[1] Dasgupta I, Wang J, Chiappa S, Mitrovic J, Ortega P, Raposo D, Hughes E, Battaglia P, Botvinick M, Kurth-Nelson Z. Causal reasoning from meta-reinforcement learning. arXiv preprint arXiv:1901.08162. 2019 Jan 23.

[2] Dasgupta I, Kurth-Nelson Z, Chiappa S, Mitrovic J, Ortega P, Hughes E, Botvinick M, Wang J. Meta-reinforcement learning of causal strategies. InThe Meta-Learning Workshop at the Neural Information Processing Systems (NeurIPS) 2019.

**Details Of Ethics Concerns:**

Not disclosing significant LLM usage.

---

> ### Author Response · Authors · 2025-11-20
>
> We thank you for your thoughtful feedback. We appreciate your acknowledgment of the significance of our work and address your concerns in the sequel.
>
> > **W1** “The entire study is conducted under discrete and finite CMDP settings. Consequently, both theoretical formulation and empirical validation are confined to simplified, low-dimensional environments. While the framework is theoretically sound, its applicability and scalability to high-dimensional continuous control tasks remain unverified.”
>
> First, we acknowledge that addressing computational challenges in high-dimensional domains is a critical issue when learning meta-RL policies. This paper explores an additional aspect that, while orthogonal to computational challenges, is equally significant: the challenge of confounding bias. We propose the first systematic approach to learning a robust policy initialization from confounded observations that generalizes well to other heterogeneous environments. The proposed algorithm is applicable to confounded MDPs with discrete state space, which cover a large number of real-world applications. We also recognize that developing confounding-robust meta-RL in MDPs featuring complex, continuous state space represents an important research direction that we are actively pursuing.
>
> > **W2** “The literature review could be more comprehensive. While the paper states that research on handling unobserved confounders in meta-reinforcement learning is still missing, several existing works have already investigated this direction using causal approaches.”
>
> Thank you for the references. While Dasgupta et al. (2019a, 2019b) studied the application of meta-RL in causal inference, their focus was primarily on the causal discovery problem. This problem aims to recover the graphical structure of a finite number of observed variables (five or fewer than 5). This focus is different from the setting addressed in our work, where we aim to learn a robust policy initialization from confounded data in unknown MDPs that have infinite horizons. We have included the references in the updated manuscript (Page 2, highlighted in blue), and we invite the reviewer to check whether we have adequately addressed their concerns.

---

> > ### Author Response · Authors · 2025-11-20
> >
> > > **Q1** “Not disclosing significant LLM usage.”
> >
> >
> > We would like to clarify that we did not use any large language model for generating technical content, experiments, or analysis in the paper. The only usage of an LLM was for minor proofreading assistance (e.g., checking typos), which does not influence the scientific contribution or the substance of the manuscript.
> >
> > > **Q2** “In lines 086–090, the paper identifies the lack of “a systematic approach for performing meta-learning across heterogeneous domains with the presence of unmeasured confounding.” However, several recent studies [1,2] have already explored causal approaches to address unobserved confounders in meta-reinforcement learning. Therefore, the gap and motivation would be stronger if the authors explicitly acknowledge these prior works and clarify how their method fundamentally differs from, or advances beyond, existing causal meta-RL approaches.”
> >
> > As mentioned in W2, Dasgupta et al. (2019a, 2019b) explored the intersection of causal inference and meta-RL; however, their problem setting is distinct from the one considered in this work. Specifically, Dasgupta et al. focused on applying meta-RL methods to learn candidate causal graphs over a finite number of observed variables and used these learned graphs to enhance causal predictions. In contrast, this paper addresses the policy learning setting, where we applied partial causal identification techniques to develop a robust sequential decision-making strategy that generalizes across heterogeneous MDP environments with an infinite horizon. We have included references for Dasgupta et al. (2019a, 2019b) and revised Lines 86-90 to read: “a systematic approach for performing meta-learning in sequential decision-making with the presence of unmeasured confounding over an infinite horizon.” We invite the reviewer to check the updated manuscript to see if it addresses their concerns.
> >
> > > **Q3** “Minor typos: ‘The shorter purple route’ should be ‘The … orange …’ in line 209.”
> >
> > Thank you for your suggestions. We have corrected the typos and would invite the reviewers to review the updated manuscript.
> >
> > > **Q4** “The paper does not include comparisons with existing meta reinforcement learning based on casual methods.”
> >
> > As discussed in W2 and Q2, existing methods, such as those proposed by Dasgupta et al. (2019a, 2019b), focus on discovering causal relationships in network structures involving a limited number of observed variables. However, these methods are not suitable for the MDP settings we consider in this paper, particularly those with an infinite decision horizon. To our knowledge, this paper is the first to systematically explore meta-RL in sequential decision-making using confounded observations collected from heterogeneous MDP environments. Our proposed method consistently outperformed popular baselines in this context, including MAML and pre-trained PPO.

---

### Official Review · Reviewer_nyXi · 2025-11-01

**Soundness:** 2
**Presentation:** 2
**Contribution:** 2
**Rating:** 4
**Confidence:** 3

**Summary:**

This paper aims to address a critical and under-explored issue in meta-reinforcement learning: how to learn a policy that can robustly and quickly adapt to new tasks when the expert data used for meta-learning is contaminated by unobserved confounding variables. The authors propose a framework called Causal MAML.

This framework employs variational inference to learn a causal generative model for inferring the posterior distribution of confounding variables from biased observational data. Then, this distribution is used to generate counterfactual trajectories to "purify" the data to help the training of MAML.

The paper provides theoretical analysis to support the unbiasedness of its meta-gradient estimation and validates the effectiveness of the method through experiments in two custom-designed confounding environments (Windy Gridworld and Causal Pendulum).

**Strengths:**

1. The paper's primary strength is its rigorous formulation of the problem. It moves beyond heuristics to prove that the core issue is a biased meta-gradient resulting from confounding. The central theoretical result—that a gradient computed on ideal counterfactual data is an unbiased estimator of the true, unconfounded gradient  (Theorem 4.1)— provides a strong guarantee on the correctness of the optimization objective. This establishes a clear and principled target for the algorithm.
2. This paper focuses on the challenge of confounding robustness in Meta-RL, a highly relevant yet often overlooked issue in real-world applications. The introduction of causal inference into Meta-RL sounds interesting.

**Weaknesses:**

1. The proposed practical algorithm (Algorithm 1) is built on a foundation that is not scalable. The core "Counterfactual Bootstrap" step requires MCMC sampling from the posterior over all possible MDPs (ρb(M | Di_obs)). This is computationally infeasible for any non-trivial environment. The method's success in the paper is an artifact of using toy domains where this step is barely possible. This reliance on an unscalable sampling procedure makes the proposed algorithm impractical for real-world application.
2. The Windy Gridworld and Causal Pendulum used in the paper are essentially "toy problems", characterized by low-dimensional state spaces and simple dynamics models. While these environments help illustrate the concept of "confounding" intuitively, they are far removed from the complexity of real-world problems.
3. The most concerning shortcoming of the paper is the complete absence of any direct evidence demonstrating that its causal inference module actually learns meaningful information about the confounder. The paper's core claim is that it "infers and utilizes the confounding variable $U$," yet the experimental section only reports final task rewards without any analysis or visualization of the learned latent variable $U$.
In a controlled environment where the ground truth of the confounding variable is known, such validation is straightforward and necessary. For instance, the authors should visualize the relationship between $U$ and the true confounder. In the absence of such validation, the causal inference module becomes an uninterpretable "black box." We cannot determine whether the performance improvement stems from successful causal inference or merely because the VAE structure happened to learn a useful—but causally irrelevant—abstract representation in these simple tasks. This significantly undermines confidence in the methodological core contribution.
4. The comparisons in the paper are limited to standard MAML and a simple pre-training baseline. This overlooks a substantial body of related work in unsupervised/self-supervised RL, which focuses on learning latent representations or skills from reward-free interactions (e.g., SMM, DIAYN). These methods are typically evaluated on more complex and widely adopted benchmarks (e.g., DeepMind Control Suite, MuJoCo). The failure to compare Causal MAML against such methods, or at least test it in environments of comparable complexity, makes its contribution appear somewhat isolated and raises serious doubts about its scalability.

**Questions:**

1. Could you provide a qualitative analysis to demonstrate that the learned latent variable $U$ indeed captures the true confounding information? For instance, by visualizing the relationship between $U$ and the actual wind force/spring stiffness.
2. How would your method handle confounding variables that dynamically change (non-stationary) or affect system dynamics in non-additive ways? Is the current framework sufficient to address these, or would it require significant modifications?
3. Have you considered evaluating your method on more challenging benchmarks (such as task sets with introduced confounding variables in MuJoCo) to demonstrate its scalability? Compared to self-supervised RL methods designed to learn latent environmental factors, what advantages does your approach offer?

---

> ### Author Response · Authors · 2025-11-20
>
> We are grateful for your detailed comments and for acknowledging our work’s importance and novelty. We have addressed your questions in the sequel.
>
> > **W1** “The proposed practical algorithm (Algorithm 1) is built on a foundation that is not scalable. The core "Counterfactual Bootstrap" step requires MCMC sampling from the posterior over all possible MDPs (ρb(M | Di_obs)). This is computationally infeasible for any non-trivial environment. The method's success in the paper is an artifact of using toy domains where this step is barely possible. This reliance on an unscalable sampling procedure makes the proposed algorithm impractical for real-world application.”
>
> We utilize the Counterfactual Bootstrap method in our algorithm, as it provides a convergence guarantee outlined in Theorem 1. For more complex environments, we offer a closed-form solution for the bounds related to potential transition probabilities in Appendix A. In this scenario, we can approximate the posterior sampling using uniform samples drawn from these bounds. In fact, our experiments follow this approximation procedure.
>
> > **W2** “The Windy Gridworld and Causal Pendulum used in the paper are essentially "toy problems", characterized by low-dimensional state spaces and simple dynamics models. While these environments help illustrate the concept of "confounding" intuitively, they are far removed from the complexity of real-world problems.”
>
> First, we would like to clarify that we do not have an environment named "Causal Pendulum." Could the reviewer please provide more details regarding this point? To the best of our knowledge, this work is the first to systematically address the challenges of unobserved confounding in meta-RL for MDP environments with discrete states. We evaluate our proposed algorithm in MiniGrid environments, which are well-known benchmarks for illustrating the challenges posed by confounding bias in discrete states (Kallus & Zhou, NeurIPS 2020; Zhang & Bareinboim, UAI 2025). We acknowledge that computation in high-dimensional domains presents a significant challenge for meta-RL. However, we believe that the issues of confounding bias are equally important and have not been adequately addressed by the existing literature. Our intention is to establish an initial foundation and initiate a broader discussion within the community, which future work can build upon to extend into more complex and high-dimensional domains.
>
> > **W3** “The most concerning shortcoming of the paper is the complete absence of any direct evidence demonstrating that its causal inference module actually learns meaningful information about the confounder. The paper's core claim is that it "infers and utilizes the confounding variable ," yet the experimental section only reports final task rewards without any analysis or visualization of the learned latent variable . In a controlled environment where the ground truth of the confounding variable is known, such validation is straightforward and necessary. For instance, the authors should visualize the relationship between  and the true confounder. In the absence of such validation, the causal inference module becomes an uninterpretable "black box." We cannot determine whether the performance improvement stems from successful causal inference or merely because the VAE structure happened to learn a useful—but causally irrelevant—abstract representation in these simple tasks. This significantly undermines confidence in the methodological core contribution.”
>
> First, our proposed counterfactual bootstrap does not utilize VAE structure. Could the reviewer elaborate on what they mean by “We cannot determine whether the performance improvement stems from successful causal inference or merely because the VAE structure happened to learn a useful—but causally irrelevant—abstract representation in these simple tasks?”
>
> The goal of causal inference is not to fully recover unobserved confounders, as their precise values cannot be determined from confounded observations. Instead, causal inference methods typically aim to infer partial information about interventional distributions and the causal effects of actions, such as transition probabilities and reward functions in MDPs. The theoretical guarantee provided in Corollary 1 ensures that our counterfactual bootstrap can construct an equivalence class of potential transition probabilities that is consistent with the observed data. In canonical models, the latent states do not represent the actual unobserved confounders; rather, they serve as sufficient statistics that allow us to recover causal effects. Additionally, we have included a closed-form characterization of the set of potential transition probabilities in Appendix A. The proof follows the classic result established by Manski in 1990.

---

> > ### Author Response · Authors · 2025-11-20
> >
> > > **W4** “The comparisons in the paper are limited to standard MAML and a simple pre-training baseline. This overlooks a substantial body of related work in unsupervised/self-supervised RL, which focuses on learning latent representations or skills from reward-free interactions (e.g., SMM, DIAYN). These methods are typically evaluated on more complex and widely adopted benchmarks (e.g., DeepMind Control Suite, MuJoCo). The failure to compare Causal MAML against such methods, or at least test it in environments of comparable complexity, makes its contribution appear somewhat isolated and raises serious doubts about its scalability.”
> >
> > We respectfully argue that SMM and DIAYN are methods focused on representation learning that tackle challenges distinct from those of meta-RL. Both SMM and DIAYN assume that the environment's dynamics remain stable, allowing them to learn latent representations of the system dynamics that can benefit future control tasks within the same environment. In contrast, meta-RL algorithms do not generally rely on these strong invariance assumptions, as system dynamics can vary significantly between training and testing environments, such as differences in wind strength or maze layout.
> >
> > Our paper introduces a confounding-robust meta-RL framework that is fundamentally compatible with many existing meta-RL and representation-learning methods. The two variants we present—Causal MAML and causal pretrained-PPO—are illustrative examples rather than an exhaustive list of baselines. To the best of our knowledge, this is the first work to investigate confounding in the context of meta-RL, and our aim is to lay a groundwork that future research can build upon to explore richer benchmarks and a wider array of unsupervised and self-supervised reinforcement learning algorithms. We also acknowledge that combining invariance assumptions with meta-RL represents a promising future research direction.
> >
> > > **Q1** “Could you provide a qualitative analysis to demonstrate that the learned latent variable  indeed captures the true confounding information? For instance, by visualizing the relationship between  and the actual wind force/spring stiffness.”
> >
> > The purpose of the counterfactual bootstrap, as explained in the response to W3, is not to directly recover or estimate the true confounding variable. Instead, similar to previous work on partial identification, we have obtained an equivalence class of possible transition probabilities and reward functions that are consistent with the confounded observations. The validity of the counterfactual bootstrap is supported by both theoretical and empirical evidence. This includes the equivalence result stated in Corollary 1, the closed-form bounds detailed in Appendix A, and the consistent performance improvements of Causal MAML observed in Experiments 1-3.
> >
> > > **Q2** “How would your method handle confounding variables that dynamically change (non-stationary) or affect system dynamics in non-additive ways? Is the current framework sufficient to address these, or would it require significant modifications?”
> >
> > Could the reviewer please elaborate on what is meant by “non-additive ways”? This paper does not impose any restrictions on the unobserved confounders, apart from the Markov property concerning the observed variables, which aligns with much of the existing reinforcement learning literature. The values of unobserved confounders can take on complex forms and follow arbitrary distributions. For example, in the Windy Gridworld example, the wind distributions are state-dependent and vary across different grids. We believe that our formulation effectively addresses the complexities of unobserved confounders in MDP environments.
> >
> > > **Q3** “Have you considered evaluating your method on more challenging benchmarks (such as task sets with introduced confounding variables in MuJoCo) to demonstrate its scalability? Compared to self-supervised RL methods designed to learn latent environmental factors, what advantages does your approach offer?”
> >
> > As explained in the responses to W2 and W4, this paper is the first to systematically investigate the challenges posed by unobserved confounding in meta-RL for sequential decision-making. As a result, we limit the scope of this work to standard MDPs with discrete observed states and unobserved confounders. However, we do expect the closed-form bounds in Appendix A could scale to high-dimensional settings, and could be consistently estimated using function approximation such as CNNs.
> > As explained in the response to W4, self-supervised RL is a specific class of pre-training methods that requires the latent representation to remain invariant across training and testing environments. This is orthogonal to the meta-RL setting considered in this work, which allows for significant shifts in the underlying system dynamics.

---

> ### Comment · Reviewer_nyXi · 2025-11-25
> **Response by Reviewer nyXi**
>
> Thank you for the detailed responses and for clarifying the point regarding the “Causal Pendulum” environment. I understand now that this is not intended to be a standard benchmark but rather a custom-designed illustrative environment to highlight the confounding structure. I appreciate the additional explanation and the motivation behind choosing these controlled settings.
>
> After carefully reading the rebuttal and revisiting the paper, I find that several of my original concerns remain. While I recognize the value of using simplified environments to study specific causal phenomena, the current experimental setup still appears somewhat limited in its ability to support the broader claims made in the paper. In particular, a more thorough qualitative examination of the learned latent variables, along with demonstrations in environments of higher complexity, would substantially strengthen the empirical foundation of the work.
>
> Overall, I continue to believe the paper presents interesting ideas and a promising theoretical direction. However, given the current level of empirical validation, I will maintain my original score. I encourage the authors to further develop the experimental section in future revisions, as the framework has the potential to make a meaningful contribution with additional evidence.

---

> > ### Author Response · Authors · 2025-11-27
> >
> > We thank the reviewer for their continued engagement in this discussion.
> >
> > > “Thank you for the detailed responses and for clarifying the point regarding the “Causal Pendulum” environment. I understand now that this is not intended to be a standard benchmark but rather a custom-designed illustrative environment to highlight the confounding structure. I appreciate the additional explanation and the motivation behind choosing these controlled settings.”
> >
> > We find the ongoing discussion about the "Causal Pendulum" somewhat curious. To the best of our knowledge, this work does not utilize any "Causal Pendulum" environment. Could you please direct us to the specific references to this environment in the submitted manuscript?
> >
> > Furthermore, in your original review, you stated, "We cannot determine whether the performance improvement stems from successful causal inference or merely because the VAE structure happened to learn a useful representation." This comment is surprising since the proposed method does not employ a VAE framework. Could you please point out the specific references to "the VAE structure" in the manuscript or explain why you believe we utilized this structure? Clarifying these issues would help advance the discussion.
> >
> > > “While I recognize the value of using simplified environments to study specific causal phenomena, the current experimental setup still appears somewhat limited in its ability to support the broader claims made in the paper. In particular, a more thorough qualitative examination of the learned latent variables, along with demonstrations in environments of higher complexity, would substantially strengthen the empirical foundation of the work.”
> >
> > There may be some confusion between representation learning and causal inference. Learning a latent representation and examining its properties in relation to the ground truth are standard practices in representation learning. However, as we mentioned in our original response, "the goal of causal inference is not to fully recover unobserved confounders," since it is widely recognized that unobserved confounders are non-identifiable (Pearl, 2000)—their precise values cannot be determined from confounded observations. In contrast, the causal effects of actions can be identified or partially identified (as is the case in this work). In this paper, the latent states in canonical causal models serve merely as sufficient statistics for computing unknown causal effects. The qualitative examination should focus on the estimated causal effects, as they are the primary target of inference. Our experiments clearly support the broader claims made in this paper, as they demonstrate that the proposed approach can reconstruct the set of candidate causal effects from observational data and utilize these estimations to warm-start future decision-making tasks, despite the presence of confounding bias.
> >
> > Finally, we acknowledge that examining and reparameterizing the latent states in canonical causal models into more meaningful and interpretable formats has been studied under the rubric of causal representation learning, which is an exciting direction for future research.

---

### Official Review · Reviewer_VhtH · 2025-11-07

**Soundness:** 3
**Presentation:** 3
**Contribution:** 3
**Rating:** 4
**Confidence:** 4

**Summary:**

The authors propose a robust meta learning method that can perform effectively in environments with unmeasurable confounding factors that affect the environment dynamics. The key idea is to use causal inference and partial identification of confounding variables to overcome this. By augmenting counterfactual trajectories from an environmental model consistent with the observed data repeatedly, the proposed method un-biases the meta learner from effects of confounding variables. The authors present in depth theoretical proofs and empirical experiments in “Windy Gridworld” environment (unobserved wind patterns as confounding factor) show that the proposed method outperforms vanilla MAML and other RL-based methods.

**Strengths:**

The paper is well structured and easy to read. The authors motivate the problem well, by clearly identifying gaps in existing meta-RL algorithms. Bringing ideas from causal inference into meta-learning applications is quite novel and the results are promising, compared to vanilla meta learning methods. The detailed theoretical analysis, well outlined pseudo algorithms, description of the experiments conducted in the grid world and the performance achieved is very interesting.

**Weaknesses:**

A key concern is the choice of baselines - while the proposed method clearly is better than vanilla MAML, I think a comparison to other state of the art distributional robust [1] or bayesian [2] meta learning methods would put the strength of the proposed algorithm in better perspective. Quantitative comparison against these more relevant baselines would have made the contributions much more impactful.

The fact that Causal PPO, In appendix C, matches or beats the proposed approach in all the tasks supports the need for stronger baselines.

Another approach to tackle confounding variables could be to formulate it as partially observed markov decision processes (POMDPs) and leverage methods like RL^2 [3]. What are some advantages of using a causal inference approach over this?

Minor typo : In appendix B.2 “log” appears twice in the equation. This is most probably a typo.

Minor : The plot colors in the main paper are not consistent with the appendix, it would be great if they are consistent. PPO is “green” in the appendix but “orange” in the main paper.

[1] A Simple Yet Effective Strategy to Robustify the Meta Learning Paradigm https://arxiv.org/abs/2310.00708
[2] https://arxiv.org/abs/1806.03836 Bayesian model agnostic meta learning
[3] RL^2 https://arxiv.org/abs/1611.02779

**Questions:**

Causal PPO outperforming the proposed causal MAML approach brings forward the question of why we need meta-learning at all? The key seems to be having counterfactual data augmentation. Do the authors have some thoughts on certain tasks where a causal MAML would hold advantage over Causal PPO?

---

> ### Author Response · Authors · 2025-11-20
>
> We thank you for your thoughtful feedback. We appreciate your acknowledgment of the significance of our work and have addressed your concerns in the sequel.
>
> > **W1** “A key concern is the choice of baselines - while the proposed method clearly is better than vanilla MAML, I think a comparison to other state of the art distributional robust [1] or bayesian [2] meta learning methods would put the strength of the proposed algorithm in better perspective. Quantitative comparison against these more relevant baselines would have made the contributions much more impactful. The fact that Causal PPO, In appendix C, matches or beats the proposed approach in all the tasks supports the need for stronger baselines. ”
>
>
> First, we would like to clarify that pre-training methods can sometimes outperform meta-learning approaches, as noted in the literature (Zhao et al., 2022; Gao & Sener, 2020). This observation applies to all meta-RL approaches that utilize expected risk minimization, such as MAML and Bayesian MAML. There is no universal consensus on whether pre-training or meta-learning is superior; the choice ultimately depends on the diversity of the available data.
>
> That said, our goal in this paper is not to settle the debate between pre-training and meta-learning. Instead, we propose a general data augmentation technique that can enhance standard few-shot RL algorithms, making them more robust to confounding biases. We have chosen MAML as our baseline because it is the most popular approach and is widely regarded as the canonical meta-RL algorithm. However, our learning strategy can be easily incorporated into other few-shot learning algorithms, and we will elaborate on this point in response to Q1.
>
> [1] Mandi Zhao, Pieter Abbeel, and Stephen James. On the effectiveness of fine-tuning versus meta-
> reinforcement learning. Advances in neural information processing systems, 35:26519–26531,
> 2022.
>
> [2] Katelyn Gao and Ozan Sener. Modeling and optimization trade-off in meta-learning. Advances in
> neural information processing systems, 33:11154–11165, 2020.
>
>
> > **W2** “Another approach to tackle confounding variables could be to formulate it as partially observed markov decision processes (POMDPs) and leverage methods like RL^2 [3]. What are some advantages of using a causal inference approach over this?”
>
> The issue of confounding bias is separate from the violation of the Markov property, which is typically found in POMDP. This indicates that standard POMDPs cannot effectively model unobserved confounding. For instance, recent research by (Li et al. NeurIPS 2025) has demonstrated that recurrent policies learned from confounded observations are suboptimal, as illustrated in Table 1. This paper aims to take the first step toward addressing robust meta-RL in sequential decision-making settings where the Markov property holds, yet unobserved confounding is present. We recognize that developing robust meta-RL techniques for confounded POMDPs with non-Markovian system dynamics and confounding bias is an essential area for future research.
>
> We also test RL^2 in our three confounding environments (Pick-Up-Key, Go-To-Door, and Go-To-Goal) and report the results in Appendix C.2. The average rewards are -0.10, -0.09, and,  -0.12 respectively, which are significantly lower than those achieved by our causal-MAML (1.21, 0.65, and 1.00) and causal pre-trained PPO (1.28, 0.82, and 1.05).
>
> [3] Confounding Robust Deep Reinforcement Learning: A Causal Approach M. Li, J. Zhang, E. Bareinboim. In Proceedings of the 39th Annual Conference on Neural Information Processing Systems, 2025
>
> > **W3** “Minor typos: 1.In appendix B.2 “log” appears twice in the equation. This is most probably a typo. 2. The plot colors in the main paper are not consistent with the appendix, it would be great if they are consistent. PPO is “green” incoin the appendix but “orange” in the main paper.”
>
> Thank you for your suggestions. We have fixed the typos in the updated manuscript. We also adjust the curve colors in Fig.7, making them consistent to the colors in Fig.6.

---

> > ### Author Response · Authors · 2025-11-20
> >
> > > **Q1** “Causal PPO outperforming the proposed causal MAML approach brings forward the question of why we need meta-learning at all? The key seems to be having counterfactual data augmentation. Do the authors have some thoughts on certain tasks where a causal MAML would hold advantage over Causal PPO?”
> >
> > As mentioned in our response to **W1**, there is an ongoing discussion about the performance comparison between pre-training and meta-RL. The choice between these approaches typically depends on the diversity of the data (Zhao et al., 2022; Gao & Sener, 2020). Our objective in this paper is not to settle this discussion, but rather to propose a novel data augmentation method that enhances the robustness of existing meta-RL algorithms against confounding bias in observational data. One advantage of our proposed approach is its immediate applicability to other few-shot learning algorithms, including pre-training, thereby enabling them to become confounding-robust as well. For instance, in Appendix C, we demonstrate how the counterfactual bootstrapping method can be used to enhance pre-training techniques, with simulations indicating that it leads to improved performance in PPO.

---

> > > ### Comment · Reviewer_VhtH · 2025-11-23
> > >
> > > Thank you for the response. Elaborating on the point that the proposed method is a general data augmentation technique rather than settling the debate between pre-training vs post-training will bring more clarity to the narrative of the paper.
> > >
> > > Also appreciate the experiments with RL^2. Its interesting how poorly this method performs, but overall addition of this baseline strengthens the paper. I would also request the authors to add a short description of why they think RL^2 performed so poorly, as they mentioned earlier that "recurrent policies learned from confounded observations are suboptimal".
> > >
> > > I will raise my score based on this response. Thanks

---

> > > > ### Author Response · Authors · 2025-11-25
> > > >
> > > > We greatly appreciate your thoughtful comments and prompt feedback. We have revised the paper to clarify that our approach functions as a general data augmentation method, specifically in lines 81, 177, and 309.
> > > >
> > > > Regarding the performance of RL^2, we believe the key issue is that recurrent policies depend solely on observation trajectories generated by a behavior policy interacting with candidate environments. Due to the presence of unobserved confounders, these observations may include spurious correlations between actions and subsequent outcomes, which hinders accurate estimation of actual causal effects, such as state-action values. In contrast, our data augmentation technique produces additional posterior counterfactual trajectories, which facilitates robust state-value estimation during policy training.
> > > >
> > > > Thank you very much for your constructive suggestions and for considering raising your score.

---

### Meta-Review · Area_Chair_7FnL · 2026-01-02

**Summary:**

The paper proposes a technically sound approach to confounding-robust meta-reinforcement learning, with theoretical development. However, its practical impact and generality remain unclear, as the evaluation is limited to simplified environments and does not convincingly demonstrate scalability or broad applicability. Several core concerns raised by reviewers, particularly regarding empirical robustness, baseline strength, and real-world relevance, remain only partially addressed after rebuttal.

**Reviewer Concerns:**

The rebuttal addressed several concerns, including additional explanations of the causal formulation, justification of design choices, minor corrections, and limited added comparisons, but the substantive concerns remain outstanding. Note reviewers’ doubts about scalability beyond toy environments, the limited empirical scope and robustness of the evaluation, the absence of strong or diverse baselines, and the lack of convincing evidence for practical impact and generalisation were not fully resolved.

**Reviewer Scores:**

All reviewers initially gave negative borderline scores. Given that the rebuttal provided only limited clarifications and cannot resolve the fundamental weakness, it is unlikely that reviewers would increase their scores, and they would most likely remain unchanged.

---

### Decision · Program_Chairs · 2026-01-26

Reject